# Alcohol attenuates CRF-induced excitatory effects from the extended amygdala to dorsostriatal cholinergic interneurons

Amanda Essoh[1†], Xueyi Xie[1†], Himanshu Gangal[1], Zhenbo Huang[1], Ruifeng Chen[1], Ziyi Li[1], Xuehua Wang[1], Valerie Vierkant[1], Miguel A Garza[1], Lierni Ugartemendia[2], Maria E Secci[3], Nicholas W Gilpin[3], Nicholas J Justice[2], Robert O Messing[4], Jun Wang[1]*

[1]Department of Neuroscience and Experimental Therapeutics, College of Medicine, Texas A&M University Health Science Center, Bryan, United States; [2]Center for Metabolic and Degenerative Disease, Institute of Molecular Medicine, University of Texas, Houston, United States; [3]Department of Physiology, Louisiana State University Health Sciences Center, New Orleans, United States; [4]Department of Neuroscience, University of Texas, Austin, United States

*For correspondence: jwang188@tamu.edu

[†]These authors contributed equally to this work

Competing interest: The authors declare that no competing interests exist.

## eLife Assessment

This **important** work shows that corticotrophin-releasing factor is delivered monosynaptically to dorsal striatal cholinergic interneurons from the central amygdala and bed nucleus of the stria terminalis. CRF increases cholinergic interneuron firing and release of acetylcholine, and this action is attenuated by pre-exposure to ethanol, suggesting a potential role in stress- and alcohol use disorders. This revision addressed prior concerns, presented **convincing** evidence supporting the conclusions, and set the stage for additional studies.

**Abstract** Alcohol relapse is associated with corticotropin-releasing factor (CRF) signaling and altered reward pathway function, though the precise mechanisms remain unclear. Here, using both mice and rats, we investigated how CRF modulates cholinergic interneurons (CINs) in the dorsal striatum, a region critical in mediating cognitive flexibility and action selection. Using monosynaptic and retrograde circuit tracing, we identified direct inputs from CRF-expressing (CRF+) neurons in the central amygdala (CeA) and bed nucleus of the stria terminalis (BNST) to dorsal striatal CINs. We showed that CINs express CRF receptor 1 (CRFR1) and established their functional connectivity with CeA/BNST CRF+ projections. Functional recordings revealed that CRF enhanced CIN excitability and promoted acetylcholine release in the dorsal striatum. However, acute alcohol exposure and withdrawal attenuated the excitatory effect of CRF on CIN firing, suggesting a mechanism by which alcohol disrupts CRF-dependent neuromodulation. These findings reveal a previously unrecognized CRF-CIN pathway linking the extended amygdala to the dorsal striatum and provide new insight into how CRF and alcohol interact to impair striatal function. This work highlights CRF signaling as a potential target for understanding stress-induced changes to the reward pathway.

**eLife digest** Stress and addiction are closely connected. Chronic stress can make the brain more sensitive to drugs and alcohol, increasing the risk of addiction. A key brain region involved in stress responses is the extended amygdala, which communicates with other areas responsible for decision-making and habit formation.

Within this system, the chemical messenger CRF triggers stress responses. Cholinergic interneurons, a specific type of neuron in the striatum, help balance brain activity and regulate motivation and behavior. These neurons also control dopamine, a chemical messenger essential for learning and reward. Although alcohol interacts with stress systems, its effects on communication between stress-related brain regions and the striatum remain poorly understood. Understanding how stress signals affect cholinergic interneurons could provide insight into addiction and other mental health conditions.

Essoh et al. aimed to investigate how alcohol influences stress signals from the extended amygdala to cholinergic interneurons in the dorsal striatum of mice and rats. Using mouse brain slices, they found that CRF released from extended amygdala inputs exerts an excitatory effect on cholinergic interneurons in the dorsal striatum. These neurons play a key role in learning and goal-directed behavior. However, alcohol significantly reduced this CRF-induced excitation.

This effect appears to result from local actions within the striatum, involving inhibition of synaptic transmission. These findings suggest that alcohol suppresses stress-related communication in brain circuits governing motivation and habit formation, which may impair the brain's ability to adapt to changing environments and promote compulsive or habitual alcohol consumption.

These insights could inform treatments for addiction and stress-related psychiatric disorders. By understanding how alcohol disrupts stress circuits, it may be possible to develop interventions that restore healthy connectivity between brain regions involved in motivation and behavioral control. Future studies in live animals are needed to confirm these effects and assess whether they produce long-term behavioral changes, particularly those associated with compulsive drug use.

## Introduction

Alcohol use disorder (AUD) is a chronic, relapsing brain condition affecting over 14 million adults in the United States, characterized by compulsive alcohol consumption, impaired control over drinking, and negative emotional states during withdrawal. Stress is a key contributor to both the development and recurrence of AUD, with stressful experiences and elevated stress hormone levels frequently precipitating relapse episodes in abstinent individuals (*Heilig and Koob, 2007*; *Becker, 2012*; *Quadros et al., 2016*). An expanding body of clinical and preclinical evidence highlights the importance of stress-responsive neurocircuits in driving addiction-related behaviors, implicating neuropeptides like corticotropin-releasing factor (CRF) in relapse vulnerability (*Bruijnzeel and Gold, 2005*; *Roberto et al., 2017*; *Koob, 1999*; *Haass-Koffler and Bartlett, 2012*; *Koob, 2008*). CRF is a central regulator of the stress response, coordinating hormonal and behavioral adaptations to stress through widespread action in both hypothalamic and extrahypothalamic brain regions. In addition to initiating hypothalamic-pituitary-adrenal axis activity, CRF modulates affective and motivational processes through its action within the amygdala, bed nucleus of the stria terminalis (BNST), and other limbic structures.

Substantial evidence indicates that CRF signaling promotes drug-seeking behavior during stress and withdrawal across multiple substances of abuse, supporting its role as a critical mediator of relapse (*Roberto et al., 2017*; *Koob, 1999*; *Shalev et al., 2010*). Despite extensive research on CRF in limbic areas, its role in the dorsal striatum, an area crucial for habit formation and behavioral flexibility, remains less well understood (*Baumgartner et al., 2021*; *Kimchi et al., 2009*; *Nonomura et al., 2018*; *Redgrave et al., 2010*; *Mantsch, 2022*; *Lemos et al., 2019*; *Carboni et al., 2018*). Within this region, cholinergic interneurons (CINs) are key regulators of striatal output and acetylcholine (ACh)-mediated modulation of dopamine signaling, integrating diverse inputs and contributing to reward-based learning (*Abudukeyoumu et al., 2019*; *Tanimura et al., 2018*; *Chantranupong et al., 2023*). CINs are sensitive to neuromodulatory influences, yet it remains unclear whether they are directly targeted by CRF and how this interaction might be altered by alcohol exposure (*Blomeley et al., 2011*; *Lim et al., 2014*; *Li et al., 2025*).

In this study, we investigated a novel CRF-CIN circuit linking the CeA and BNST to the dorsal striatum. Using monosynaptic and retrograde circuit tracing, we identified direct projections from CRF-expressing neurons to dorsal striatal CINs. CRF enhanced CIN excitability and promoted ACh release via CRFR1 activation, but this excitatory effect was disrupted by acute alcohol exposure, indicating that alcohol interferes with CRF-dependent cholinergic modulation. These findings identify a CRF-CIN circuit that is vulnerable to alcohol-induced dysregulation, providing mechanistic insight into how stress peptides and alcohol interact to impair striatal function.

## Results

### Dorsal striatal CINs receive monosynaptic inputs from CeA and BNST neurons

To investigate the connection between dorsal striatal CINs and stress-related brain regions, we examined whether CINs receive monosynaptic inputs from the CeA and the BNST (*Davis, 2006*; *Funk et al., 2006*). We used ChAT-Cre;D1-tdTomato mice to perform rabies-mediated monosynaptic retrograde tracing. In this mouse model, CINs express Cre recombinase, and dopamine D1-receptor (D1R)-expressing medium spiny neurons (D1-MSNs) are labeled with tdTomato. Including the D1-td-Tomato marker allowed us to delineate the CeA and BNST, which, unlike surrounding striatal areas, do not express D1Rs (*Lu et al., 2021*). We performed rabies-mediated retrograde monosynaptic circuit tracing (*Figure 1A*), allowing specific targeting of CINs (*Figure 1B*). In addition to labeling neurons in brain regions known to project to dorsal striatal CINs—including the striatum itself, cortex (e.g. cingulate cortex, motor cortex, somatosensory cortex), thalamus (e.g. parafascicular thalamic nucleus, centrolateral thalamic nucleus), globus pallidus, and midbrain—we were surprised to see labeled neurons in both the CeA (*Figure 1C*) and BNST (*Figure 1D*), which are well characterized as key stress-responsive nuclei. Interestingly, the BNST had a higher density of DMS-projecting neurons than the CeA (*Figure 1E*; Mann-Whitney U=4.000, ***p<0.001). These results demonstrate that dorsal striatal CINs receive direct, monosynaptic inputs from neurons in the CeA and BNST. Together, these findings highlight the anatomical connection of dorsal striatal CINs with key stress-responsive brain regions.

### The dorsal striatum lacks CRF[+] neurons unlike the CeA and BNST

We then investigated whether the dorsal striatum contains CRF-producing cells or whether CRF signaling in this region depends on CeA and BNST inputs. While previous studies have identified CRF-producing neurons in the CeA and BNST (*de Guglielmo et al., 2019*), their presence in the dorsal striatum was uncertain. Coronal brain sections from CRF-Cre;tdTomato rats revealed dense populations of CRF-tdTomato[+] neurons in the CeA (*Figure 2A*) and BNST (*Figure 2B*). Although CRF[+] axonal fibers were observed in the dorsal striatum, CRF[+] cell bodies were largely absent (*Figure 2C*). Quantitative analysis confirmed significant regional differences, indicating that the CeA and BNST contained substantially more CRF[+] neurons than the dorsal striatum (*Figure 2D*; CeA versus DS, Q=6.62, *p<0.05; BNST versus DS, Q=4.54, *p<0.05). Notably, the CeA exhibited a similar density of CRF[+] neurons to the BNST (*Figure 2D*; Q=1.40, p>0.05). The apparent lack of CRF[+] cell bodies suggests that CRF signaling in the dorsal striatum originates from CRF[+] neurons in the CeA and BNST.

### Dorsal striatum CINs receive monosynaptic CRF[+] inputs from the CeA and BNST

To test whether CRF[+] neurons in the CeA and BNST project to the dorsal striatum, we infused a retrograde adeno-associated virus (AAV) into the dorsal striatum of CRF-Cre;tdTomato rats. This strategy labels CRF[+] neurons (tdTomato[+]) that project to the dorsal striatum with GFP (*Figure 3A*). Confocal imaging revealed GFP[+] neurons in both the BNST (*Figure 3B*) and CeA (*Figure 3C*), with substantial overlap between GFP-labeled dorsal striatum-projecting neurons and tdTomato-positive (CRF[+]) cells in both regions. This indicates that CRF[+] neurons from the CeA and BNST project to the dorsal striatum. Despite a similar overall CRF[+] neuron density in the CeA and BNST (*Figure 3D*; $t_{15}$=1.06, p>0.05), the proportion of dorsal striatum-projecting CRF[+] neurons was significantly higher in the BNST than in the CeA (*Figure 3E*; $t_{15}$=–5.01, ***p<0.001). The density of dorsal striatum-projecting

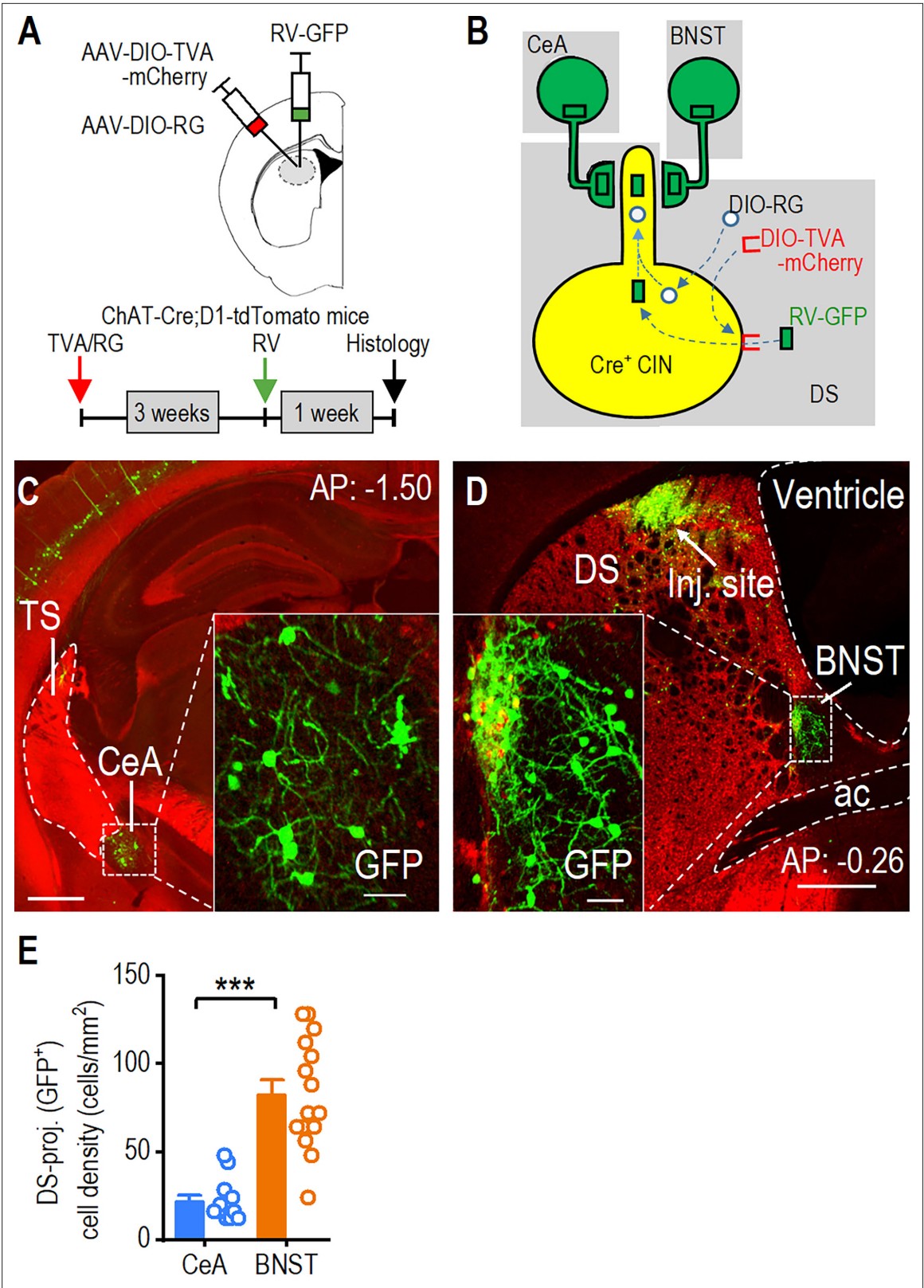

**Figure 1.** Dorsal striatal cholinergic interneurons (CINs) receive monosynaptic inputs from the central amygdala (CeA) and bed nucleus of the stria terminalis (BNST). (**A**) Schematic illustrating the infusion of helper viruses (AAV-DIO-TVA-mCherry and AAV-DIO-RG) and rabies-GFP (RV-GFP) into the dorsal striatum of ChAT-Cre;D1-tdTomato mice. The rabies-GFP was infused 3 weeks after the helper virus infusion, and animals were euthanized 1 week after the rabies infusion. (**B**) Model for the retrograde transsynaptic labeling of CeA and BNST neurons projecting to dorsal striatal CINs. TVA permits

*Figure 1 continued on next page*

*Figure 1 continued*

selective infection by the rabies-GFP virus, while RG mediates the retrograde transsynaptic jump from postsynaptic CINs to presynaptic terminals of CeA or BNST neurons. (**C**) Sample image showing CeA neurons projecting to dorsal striatal CINs, as indicated by rabies-GFP expression. AP: –1.50 mm from bregma. TS, tail of the striatum. Scale bar: 0.5 mm, 50 µm for insert. (**D**) Sample image demonstrating that BNST neurons project to dorsal striatal CINs, as indicated by rabies-GFP expression. Note that the injection site (inj. site) is also displayed in the posterior dorsal striatum. AP: –0.26 mm from bregma. DS, dorsal striatum; ac, anterior commissure. Scale bar: 0.5 mm, 50 µm for insert. (**E**) The CeA and BNST both send projections to the dorsal striatum, with the BNST providing more inputs. ***p<0.001 by Mann-Whitney test. n=13 sections from 4 mice (13/4) for the CeA and 15/4 for the BNST. Data are presented as mean ± SEM.

CRF[+] neurons was also higher in the BNST than the CeA (*Figure 3F*; $t_{15}$=–3.07, **p<0.01). These results indicate that CRF[+] neurons in the CeA and BNST project to the dorsal striatum.

To test whether CRF[+] neurons in the CeA and BNST make monosynaptic projections to dorsal striatal CINs, we performed opsin-assisted functional circuit tracing. Given that both the CeA and BNST are primarily GABAergic nuclei (*Partridge et al., 2016*; *Sanford et al., 2017*), we examined whether optogenetic stimulation of CRF-containing projections to the dorsal striatum induces inhibitory monosynaptic currents in CINs. To test this, we infused AAV-FLEX-Chrimson-tdTomato into the CeA and BNST of CRF-Cre;ChAT-eGFP mice so that we could selectively stimulate CRF-containing fibers in the dorsal striatum while recording from GFP-labeled CINs (*Figure 3G*). We found that optogenetic stimulation of CRF[+] fibers with yellow light evoked fast synaptic currents (within ~6.5 ms from the stimulation onset) in dorsal striatal CINs (*Figure 3H–J*). This response was abolished by tetrodotoxin (TTX) (*Figure 3H and I*; $\chi^2$(3)=21.9, p<0.001; BL vs. TTX: z=3.10, *p<0.05) but restored using

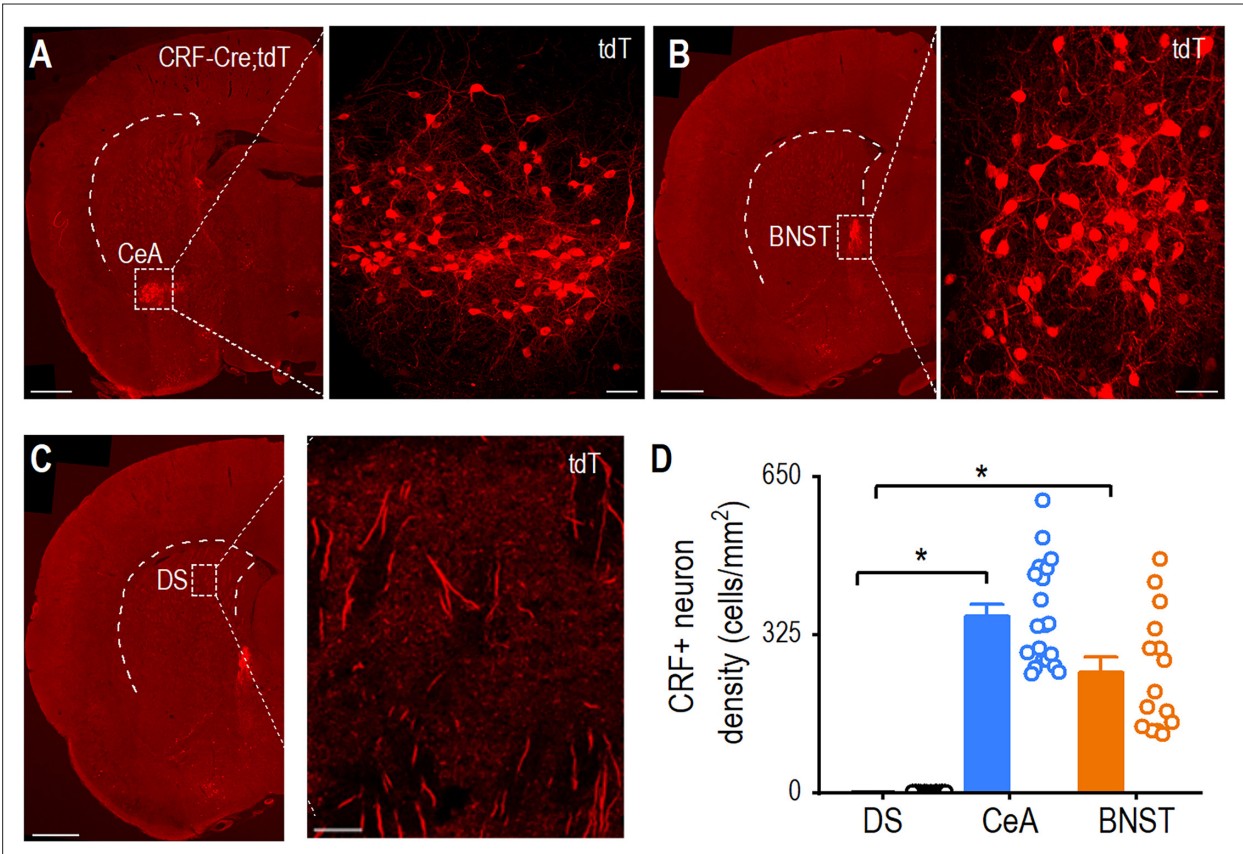

**Figure 2.** Corticotropin-releasing factor (CRF)-positive neurons are abundant in the central amygdala (CeA) and bed nucleus of the stria terminalis (BNST) but absent in the dorsal striatum. (**A, B**) Confocal images showing dense populations of tdTomato-labeled CRF[+] neurons in the CeA (**A**) and BNST (**B**) of CRF-Cre;tdTomato rats. AP: –1.5 mm from bregma (**A**) and –0.26 mm (**B**). Scale bars: 1 mm (left) and 50 µm (right). DS, dorsal striatum. (**C**) Representative image of the posterior dorsal striatum showing the presence of CRF[+] axonal fibers but the absence of CRF[+] cell bodies. AP: –0.26 mm from bregma. Scale bar: 1 mm (left) and 100 µm (right). (**D**) The CeA and BNST contain more CRF[+] neurons than the dorsal striatum. *p<0.05 by Kruskal-Wallis with Dunn's method. n=20 slices from 7 rats (20/7) for the CeA, 29/7 for the striatum, and 15/7 for the BNST. Data are presented as mean ± SEM.

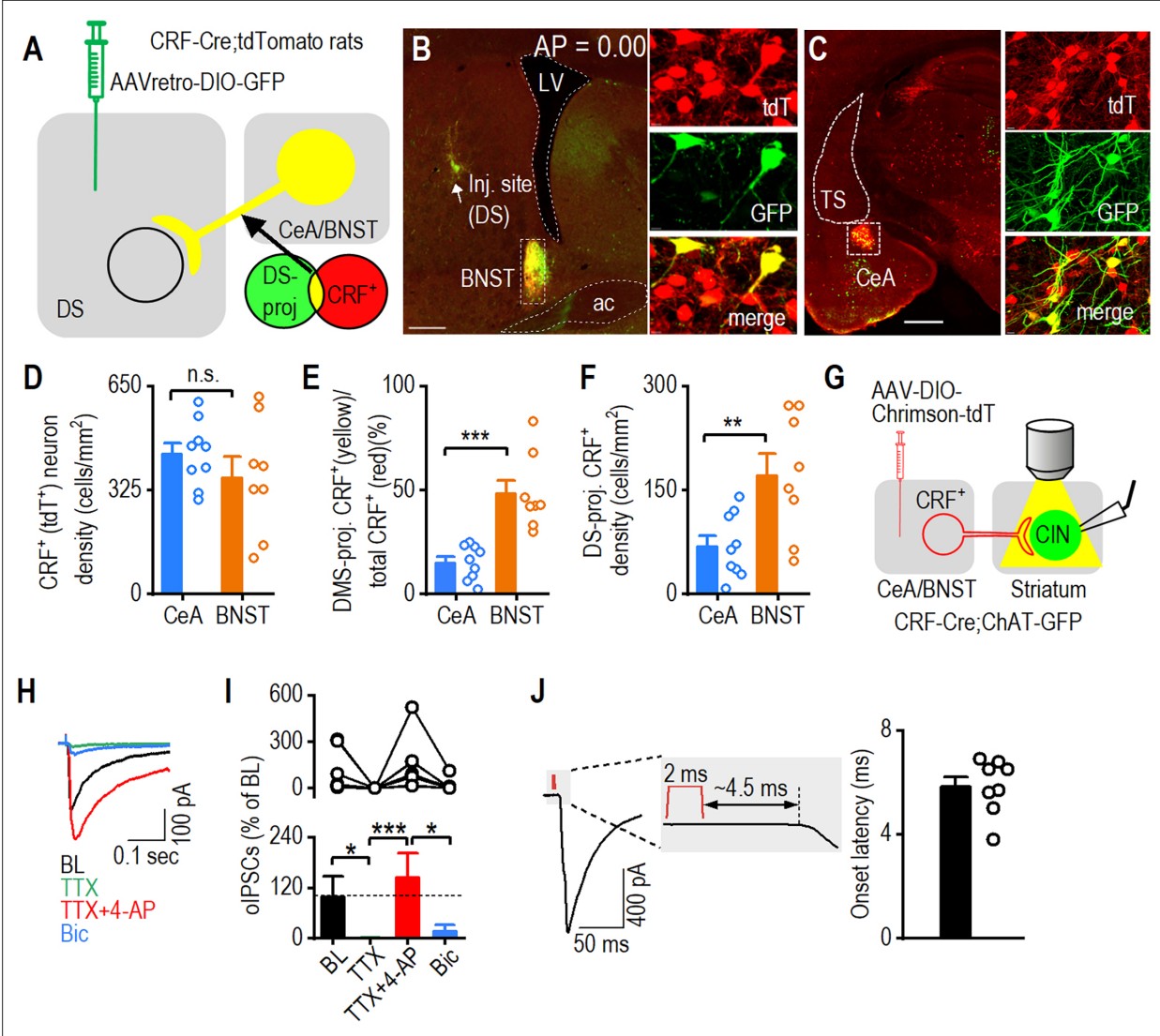

**Figure 3.** Cholinergic interneurons (CINs) receive monosynaptic input from central amygdala (CeA) and bed nucleus of the stria terminalis (BNST) CRF+ neurons. (**A**) Schematic showing virus injection (AAVretro-DIO-GFP) into the dorsal striatum of CRF-Cre;tdTomato rats. (**B**) Representative image showing the injection site of AAVretro-DIO-GFP in the posterior dorsal striatum of CRF-Cre;tdTomato rats. Overlapping tdTomato+ (CRF+) and dorsal striatum-derived Cre-driven GFP+ expression in the BNST (depicted in yellow) indicates that these CRF+ neurons project to the dorsal striatum. ac, anterior commissure; LV, left ventricle; tdT, tdTomato; DS-proj, dorsal-striatum-projecting. Scale bar: 0.5 mm (left), 10 µm (right). The AP coordinates for the injection site from each of the 5 rats are as follows: AP: 0.00 mm, 0.10 mm, 0.12 mm, 0.00 mm, 0.11 mm. (**C**) Images showing the overlap (in yellow) of tdTomato+ (CRF+) and dorsal striatum-derived Cre-driven GFP+ expression in the CeA, confirming that these CRF+ neurons also project to the dorsal striatum. TS, the tail of the striatum. Scale bar: 1 mm (left), 10 µm (right). (**D**) The CeA and BNST contain a similar CRF+ neuron density. n.s., not significant, $p > 0.05$ by unpaired t-test, n=9 sections from 5 rats (9/5) for the CeA and 8/5 for the BNST. (**E**) The proportion of CRF+ neurons projecting to the dorsal striatum is higher in the BNST compared to the CeA. ***$p < 0.001$ by unpaired t-test, n=9/5 (CeA) and 8/5 (BNST). (**F**) The density of dorsal striatum-projecting CRF+ neurons is greater in the BNST than the CeA. **$p < 0.01$. n=9/5 (CeA) and 8/5 (BNST). (**G**) Schematic showing the injection of AAV-FLEX-Chrimson-tdTomato into the CeA and BNST of CRF-Cre;ChAT-eGFP mice and the subsequent recording of green striatal CINs during blue light stimulation of surrounding Chrimson-containing fibers from CRF+ neurons. Fibers were stimulated at a wavelength of 590 nm for 2 ms. (**H**) Sample traces showing CIN responses to blue light stimulation of CRF fibers, which were abolished by tetrodotoxin (TTX), recovered with TTX+4-aminopyridine (4-AP) and further eliminated by bicuculline (Bic). (**I**) Summary of oIPSC data showing the disappearance of oIPSCs with TTX and reappearance with TTX and 4-AP. *$p < 0.05$ by one-way ANOVA. n=8 cells from 3 mice. (**J**) The average latency between the start of optogenetic stimulation and postsynaptic response was ~6.5 ms. n=8 cells from 3 mice. Data are presented as mean ± SEM.

TTX+4-aminopyridine (4-AP) (*Figure 3H and I*; TTX vs. TTX+4-AP: z=4.26, ***p<0.001), suggesting the existence of monosynaptic transmission. Notably, the restored synaptic response was predominantly blocked by the GABA$_A$ receptor antagonist bicuculline (*Figure 3H and I*; TTX+4-AP vs. TTX+4-AP+Bic: z=3.10, *p<0.05), demonstrating that evoked currents are GABAergic. Together, these results demonstrate that dorsal striatal CINs receive monosynaptic GABAergic inputs from CRF$^+$ neurons in the CeA and BNST, suggesting that these neurons project to the dorsal striatum and may release CRF to modulate cholinergic signaling.

## Striatal CINs express CRFR1

Having established that CINs receive monosynaptic inputs from CRF$^+$ neurons in the CeA and BNST, we next sought to confirm that CINs express the CRF receptor CRFR1 (*Lemos et al., 2019*; *Chen, 2016*; *Olson et al., 2024*). To address this, we used CRFR1-Cre-tdTomato rats (*Weera et al., 2022*), in which CRFR1-expressing neurons express Cre recombinase and are visualized via tdTomato fluorescence. Confocal imaging of striatal sections from these rats confirmed that CRFR1-expressing neurons are distributed throughout the striatum (*Figure 4A*). To verify that CRFR1-positive (CRFR1$^+$) neurons are also cholinergic, we stained sections with an anti-ChAT antibody and acquired confocal images (*Figure 4A*). We found that ~25% of rat CINs express CRFR1 (*Figure 4B and C*; Mann-Whitney U=0.00, ***p<0.001). To determine whether this expression pattern is conserved across species, we examined CRFR1-GFP mice and performed ChAT immunostaining on striatal sections (*Figure 4D*). We found that approximately 10% of CINs expressed CRFR1 (*Figure 4E*), indicating that while the proportion is lower than in rats, a subset of CINs in both species express CRFR1. These findings align with previous reports showing that a subset of striatal CINs express CRFR1, supporting a role for CRFR1-mediated signaling in the striatum.

## CRF enhances CIN activity and ACh release in the dorsal striatum

Following the finding that CRF$^+$ neurons send inputs to the dorsostriatal CINs, we next investigated how CRF influences spontaneous CIN firing activity. Cell-attached recordings of GFP$^+$ neurons in the dorsal striatum were conducted in ChAT-eGFP mice before and during bath application of CRF (100 nM) (*Figure 5A*). We found that CRF significantly increased CIN firing frequency (*Figure 5B and C*; $t_{(8)} = -24.08$, ***p<0.001). This effect was abolished by pretreatment with the CRFR1 antagonist NBI 35695 (*Figure 5C*; Mann-Whitney U=0.00, ***p<0.001), confirming that CRF potentiates CIN firing via CRFR1 activation in the dorsal striatum.

Because CINs are the primary source of ACh in the dorsal striatum, we next examined whether CRF-induced CIN activation modulates ACh release. To do so, we infused an AAV introducing a genetically encoded ACh sensor (AAV-hSyn-GRAB$_{ACh4m}$) into the dorsal striatum of wild-type mice (*Purvines et al., 2025*; *Huang et al., 2024*; *Touponse et al., 2025*; *Potjer et al., 2025*; *Gangal et al., 2025*; *Jing et al., 2020*). Live-tissue confocal imaging was performed on brain slices containing the dorsal striatum 14 days post-infusion to monitor ACh levels (*Figure 5D*). Bath application of CRF produced a robust ACh fluorescence signal (*Figure 5E and F*; Mann-Whitney U=7.00, *p<0.05), indicating that CRF enhances ACh release. These results suggest that CRF-driven CIN activation increases ACh release in the dorsal striatum.

To further determine whether direct activation of CRF-expressing fibers modulates CIN activity, we used an optogenetic approach. We generated CRF-Cre;Ai32;ChAT-eGFP mice to selectively stimulate CRF$^+$ terminals in the dorsal striatum. Cell-attached recordings were conducted in CINs while blue light (470 nm, 50 Hz, 60 s) activated local ChR2-expressing CRF$^+$ fibers (*Figure 5G*). Burst stimulation of CRF$^+$ terminals significantly and reversibly increased CIN firing in the presence of 6,7-dinitroquinoxaline-2,3-dione (DNQX) and bicuculline (*Figure 5H and I*; pre vs. stim, q=3.69, *p<0.05; stim vs. post, q=11.82, ***p<0.001). However, application of a CRFR1 antagonist abolished this effect (*Figure 5J and K*; $F_{(2,16)} = 0.96$, p=0.41), confirming that the increase in CIN firing was mediated by CRFR1 signaling. Together, these findings demonstrate that CRF enhances CIN activity and ACh release in the dorsal striatum.

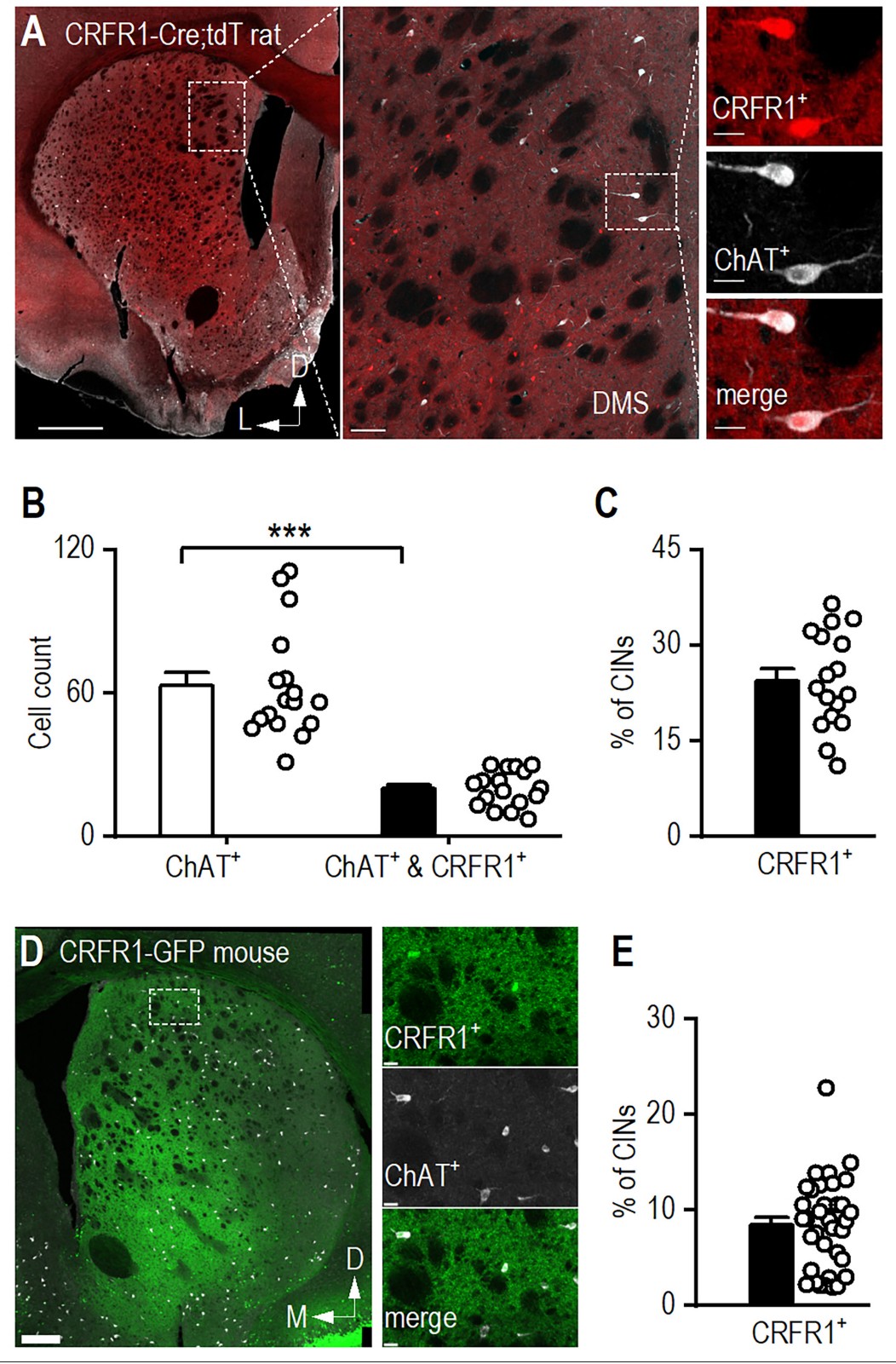

**Figure 4.** CRFR1 is expressed in striatal cholinergic interneurons (CINs). (**A**) Representative image of a coronal striatal section from a CRFR1-Cre-tdTomato rat showing CRFR1 expression overlapping with anti-ChAT immunoreactivity. The section was stained using an anti-ChAT antibody and imaged using 647-nm wavelength (white). Scale bars: 1 mm (left), 100 µm (middle), and 20 µm (right). (**B**) There are significantly fewer CRFR1+ CINs

*Figure 4 continued on next page*

*Figure 4 continued*

than the total number of CINs in the dorsal striatum. ***p<0.001 by Mann-Whitney test. n=8 sections from 2 rats. (**C**) Around 30% of CINs express CRFR1 in CRFR1-Cre;tdT rat. n=8 sections from 2 rats. (**D**) Representative image of a coronal striatal section from a CRFR1-GFP mouse showing CRFR1 expression overlapping with anti-ChAT immunoreactivity. Scale bars: 0.5 mm (left) and 20 µm (right). (**E**) Around 10% of CINs express CRFR1 in CRFR1-GFP mouse. n=35 sections from 4 mice. Data are presented as mean ± SEM.

## Acute alcohol application attenuates CRF-mediated enhancement of CIN firing

Given that acute alcohol exposure suppresses CIN activity (**Blomeley et al., 2011**), we next examined how acute alcohol application and subsequent withdrawal influence CRF-mediated enhancement of CIN firing. We first assessed the effect of acute alcohol by measuring spontaneous CIN activity before, during, and after 50 mM alcohol application in dorsal striatal slices from ChAT-eGFP mice. Consistent with previous reports (**Blomeley et al., 2011**), acute alcohol significantly suppressed CIN spontaneous firing, an effect reversed after a washout period, which we refer to here as withdrawal (**Figure 6A and B**; Mann-Whitney U=0.00, *p<0.05).

To determine how prior alcohol exposure (modeling early withdrawal) affects CRF-induced enhancement of CIN firing, we pretreated dorsal striatal slices with alcohol (50 mM) for 1 hr outside of the recording chamber (**Figure 6C**), followed by CRF application during recordings. In untreated control slices, CRF significantly increased spontaneous CIN firing, consistent with our previous findings (**Figure 6D**; BL vs. CRF, q=11.51, ***p<0.001; see also **Figure 5B**). When alcohol was subsequently applied after CRF, firing decreased but remained elevated relative to baseline (**Figure 6D**; CRF vs. EtOH, q=6.21, **p<0.01; BL vs. EtOH, q=5.29, **p<0.01), likely due to a residual CRF effect (**Figure 5B**).

In slices pretreated with alcohol (mimicking early withdrawal), CRF still increased CIN firing, but subsequent alcohol application did not fully return firing to baseline levels (**Figure 6E**; BL vs. CRF, q=5.20, **p<0.01; CRF vs. EtOH, q=6.21, p=0.05). This suggests that alcohol pretreatment (withdrawal state) attenuates CRF-induced CIN activation and diminishes alcohol's subsequent suppressive effects. Notably, when comparing the change in firing from baseline to CRF application, alcohol pretreatment blunted CRF-induced CIN activation (**Figure 6F**; $t_{(18)}$ = 2.70, *p<0.05). Together, these findings indicate that both acute alcohol exposure and early withdrawal attenuate CRF-mediated enhancement of CIN firing, supporting the conclusion that alcohol modulates striatal cholinergic signaling.

## Discussion

This study identifies a circuit in which CRF-positive neurons in the CeA and BNST provide direct input to dorsal striatal CINs that express CRFR1. CRF enhances CIN firing activity and ACh release, linking CRF to cholinergic modulation. In addition, alcohol exposure and withdrawal blunt this effect, suggesting that CRF signaling to dorsal striatal CINs may contribute to mechanisms relevant to AUD. Using monosynaptic and retrograde circuit tracing, we identified direct projections from CRF-expressing neurons in the CeA and BNST to dorsal striatal CINs. Functional electrophysiology showed that CRF enhanced CIN activity and promoted ACh release via CRFR1 receptors. However, acute alcohol exposure and withdrawal disrupted this excitatory effect, indicating that alcohol interferes with CRF-dependent cholinergic modulation. These findings identify a CRF-CIN circuit that is sensitive to alcohol-induced modulation. Striatal cholinergic signaling is essential for behavioral flexibility (**Matamales et al., 2016**), and alcohol exposure is known to impair CIN activity and ACh release (**Huang et al., 2024**; **Ma et al., 2022**), leading to reduced flexibility. Thus, alcohol-induced alterations in CRF-CIN signaling may disrupt cholinergic control of striatal circuits, providing a potential mechanism by which alcohol promotes cognitive rigidity and compulsive reward-seeking behaviors characteristic of AUD.

### Monosynaptic inputs from the CeA and BNST to CINs

Tracing experiments reported here show that dorsal striatal CINs receive direct synaptic input from CRF-expressing neurons in the CeA and BNST, regions involved in stress and emotion (**Partridge**

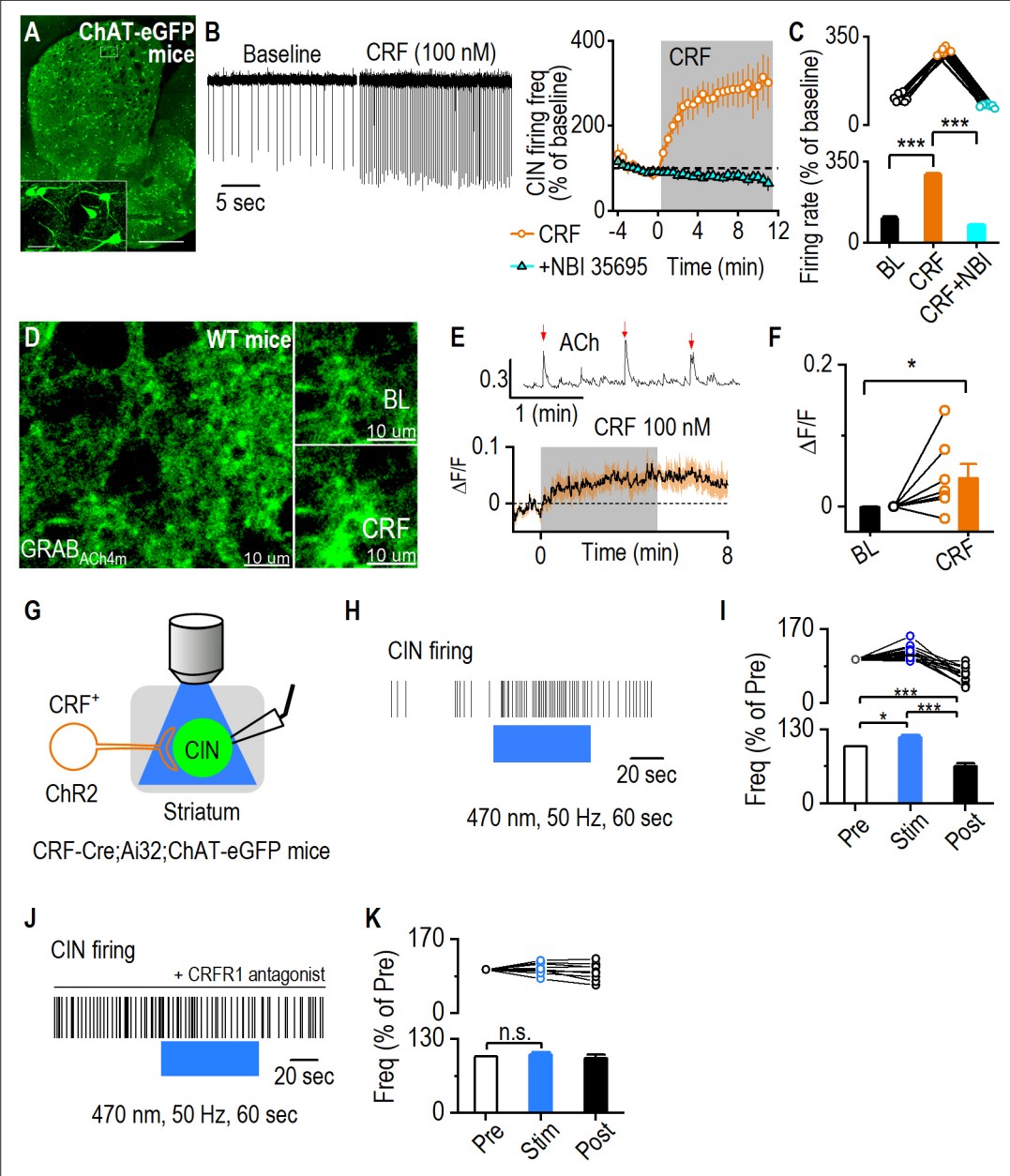

**Figure 5.** Corticotropin-releasing factor (CRF) enhances cholinergic interneuron (CIN) activity and acetylcholine (ACh) release in the dorsal striatum. (**A**) Sample images of GFP-labeled CINs in the striatum of a ChAT-eGFP mouse. Scale bar: 0.5 mm, 50 μm (inset). (**B**) Bath application of CRF (100 nM) increased the spontaneous firing of dorsal striatal CINs in cell-attached electrophysiological recordings. This effect was prevented by pretreatment with the CRFR1 antagonist NBI 35695 (5 μM). n=7 cells from 3 mice (7/3) for CRF and 6/3 for CRF plus antagonist recordings. (**C**) Data showing that CRF significantly increases the firing from baseline, and the firing frequency in the presence of CRF following pretreatment with CRFR1 antagonist is significantly lower. ***p<0.001 by paired t-test, ***p<0.001 by Mann-Whitney test. n=7 cells from 3 mice. (**D**) Sample images of ACh sensor fluorescence in dorsal striatal slices before and during bath application of CRF (100 nM). AAV-hSyn-GRAB$_{ACh4m}$ was infused into the dorsal striatum of wild-type mice, and live-tissue confocal imaging was conducted 2 weeks post-infusion. Scale bar: 10 μm for left and right. (**E**) Sample trace of spontaneous ACh release events (indicated by red arrows, top). Bath application of CRF increased ACh sensor fluorescence (bottom). (**F**) Summary data showing a significant increase in ACh sensor fluorescence following CRF application. *p<0.05 by Mann-Whitney test. n=7 slices from 7 mice. (**G**) Schematic of cell-attached electrophysiological recordings from dorsal striatal CINs in CRF-Cre;Ai32;ChAT-eGFP mice, with simultaneous optogenetic stimulation of CRF$^+$ fibers using blue light (470 nm, 2 ms, 50 Hz, 60 s). (**H**) Sample trace showing the increase in CIN firing frequency during blue light stimulation of CRF$^+$ fibers. (**I**)

*Figure 5 continued on next page*

Data demonstrating a significant and reversible increase in CIN firing frequency during optogenetic stimulation, *p<0.05, ***p<0.001 by one-way RM ANOVA. n=13 neurons from 8 animals. (**J**) Sample trace showing no change in CIN firing frequency during blue light stimulation of CRF⁺ fibers when CRFR1 antagonist (antalarmin hydrochloride, 2 μM) was bath applied. (**K**) Data demonstrating no significant change in CIN firing frequency during optogenetic stimulation. n=9 neurons from 3 mice. Data are presented as mean ± SEM.

*et al., 2016*; *Kim et al., 2013*). While previous studies have identified CeA and BNST projections to the striatum, they did not show that they arise from CRF neurons or that they target CINs (*Lu et al., 2021*; *Giovanniello et al., 2025*; *Smith et al., 2016*; *Heaton et al., 2024*). Our results support the presence of a direct CRF-to-CIN pathway, linking CRF release to cholinergic modulation. Given the role of CINs in striatal output, response flexibility, and reward learning (*Huang et al., 2024*; *Duhne et al., 2024*; *Ostlund et al., 2017*; *Dautan et al., 2020*; *Aoki et al., 2018*), this circuit likely contributes to stress-driven behavioral adaptations (*Partridge et al., 2016*; *Rieger et al., 2022*; *Alizamini et al., 2022*; *Bryce and Floresco, 2016*; *Pomrenze et al., 2019*). A limitation of our optogenetic approach is the use of CRF-Cre;Ai32 mice, in which ChR2 is expressed in all CRF⁺ neurons. Thus, although our tracing confirmed CeA and BNST inputs, we cannot exclude contributions from other CRF⁺ populations. Future work using projection-specific targeting approaches will be required to isolate CeA and BNST contributions.

Tracing and receptor expression studies were performed in mice and rats in a largely non-overlapping manner. While mice and rats share many conserved amygdalostriatal components, we did not attempt

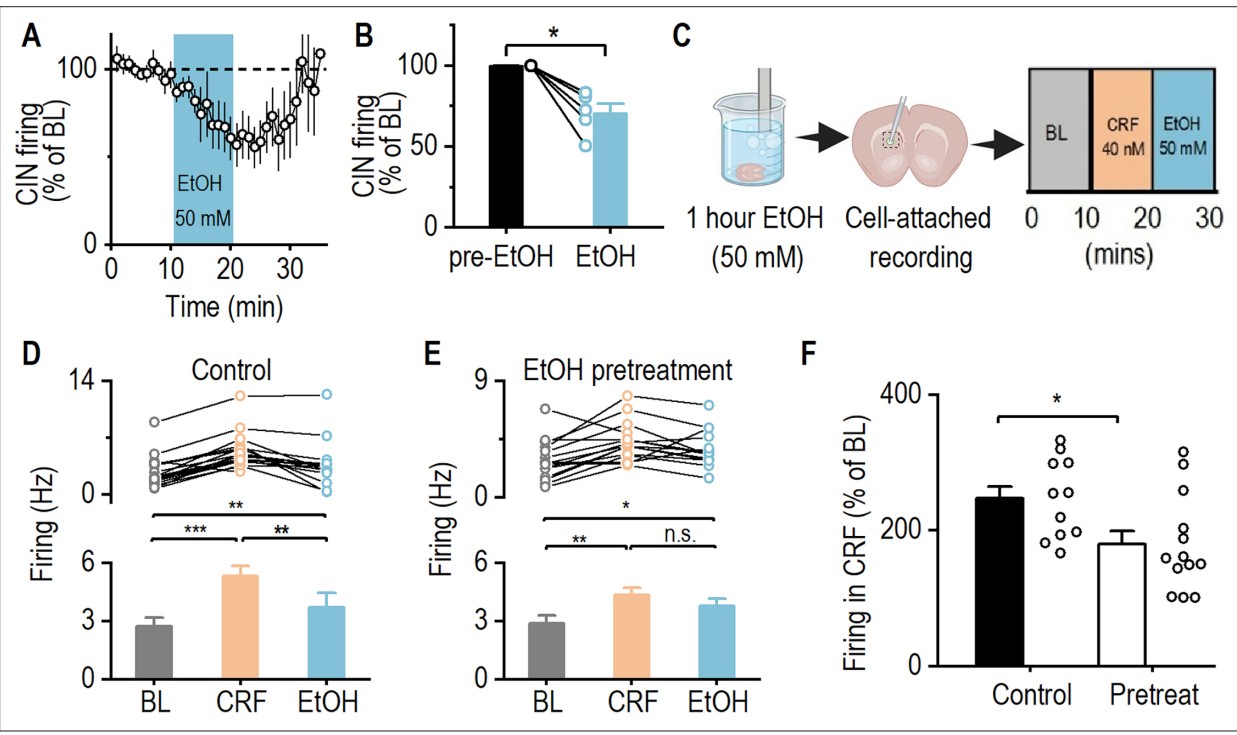

**Figure 6.** Acute alcohol exposure attenuates corticotropin-releasing factor (CRF)-induced enhancement of cholinergic interneurons (CIN) activity. (**A**) Time course of spontaneous firing in CINs from dorsal striatal slices of ChAT-eGFP mice before, during, and after bath application of alcohol (EtOH; 50 mM). (**B**) Summary data demonstrates a significant reduction in CIN firing frequency following alcohol application. **p<0.01 by Mann-Whitney test. n=6 cells from 4 mice. (**C**) Schematic of the experimental design in which striatal slices were pretreated with alcohol (50 mM) for 1 hr in an incubation chamber and then washed for 15 min in the recording chamber. CINs were selected for cell-attached electrophysiological recordings, measuring firing frequency for 10 min (baseline), followed by CRF (40 nM) and alcohol (50 mM) bath applications for 10 min each. (**D, E**) Data showing spontaneous firing of CINs during baseline (BL), CRF, and alcohol bath application for the control (**D**) and alcohol pretreated (**E**) groups. *p<0.05, **p<0.01, ***p<0.001 by one-way RM ANOVA. n=10 cells from 8 mice (10/8) for the control group and 10/7 for the alcohol pretreated group. (**F**) Data showing that alcohol pretreatment attenuated CRF-induced enhancement of CIN firing frequency. *p<0.05 by unpaired t-test. n=10 cells from 8 mice (10/8) for the control group and 10/7 for the pretreated group. Data are presented as mean ± SEM.

direct cross-species comparisons, and our findings should therefore be interpreted as species-specific. In rats, CRFR1 expression was largely restricted to a subset of CINs, consistent with previous reports. In mice, the histological distribution of CRFR1 remains less well defined; however, our recordings from GFP-labeled CINs in ChAT-GFP mice demonstrated that bath-applied CRF increased CIN firing in a CRFR1-dependent manner. These recordings were conducted in the presence of glutamatergic and GABAergic antagonists, and in some cases TTX, ensuring that CINs were functionally isolated from upstream inputs. Because striatal ACh arises almost exclusively from CINs, with only a minor contribution from brainstem cholinergic afferents, it is most likely that CRF-induced increases in ACh reflect direct CRFR1 signaling in CINs. Nevertheless, the possibility that CRFR1 is expressed in other striatal cell types remains an important open question. Another unresolved question is whether CRFR1$^+$ CINs are equally abundant across striatal subregions. While representative images may appear to suggest higher CRFR1$^+$ CIN density in the DLS compared to the DMS, we did not quantify this systematically. Future work will be needed to determine whether such regional differences exist, as they could have important implications for dorsal striatal function.

Although CRFR1 expression has been reported in a subset of striatal CINs, we found that the majority of mouse CINs increased firing in response to bath-applied CRF. Using a CRFR1-GFP reporter mouse, only ~10% of CINs showed detectable somatic CRFR1-GFP expression, indicating that widespread CIN responsiveness cannot be explained solely by direct receptor expression at the soma. Reporter lines may underestimate functional CRFR1 expression, particularly if receptors are expressed at low levels or localized to distal neuronal compartments. Consistent with a CRFR1-dependent mechanism, the CRF-induced increase in CIN firing was abolished by a selective CRFR1 antagonist. In addition, CRF may act indirectly by modulating presynaptic inputs to CINs, and electrical coupling among CINs may allow excitation of a subset of neurons to propagate across the network (*Ren et al., 2021*). Together, these mechanisms provide a parsimonious explanation for broad CIN sensitivity to CRF despite limited reporter-detected CRFR1 expression.

We found no CRF$^+$ neurons in the dorsal striatum, indicating that CRF signaling in this region originates from extrinsic inputs, including the CeA and BNST. However, CRF$^+$ neurons have been reported in the ventral striatum, where they play a role in learning-related processes (*Eckenwiler et al., 2025*). Thus, CRF's contribution to striatal function may differ along the dorsal-ventral axis, with intrinsic ventral striatal CRF$^+$ neurons complementing extrinsic dorsal striatal CRF inputs. These sources of CRF may modulate striatal circuits involved in selecting behaviors flexibly, a process critical for adaptive responses to stress. Electrophysiological experiments further show that CRF fiber stimulation enhances CIN excitability even in the presence of synaptic blockers, supporting a direct modulatory role for CRF signaling on cholinergic tone in the striatum. Moreover, the transient effects of endogenous CRF release, compared to the prolonged activation from CRF bath application, suggest that CRF signaling may generate temporally dynamic patterns of CIN activity to influence behavioral output.

## Alcohol disrupts CRF-mediated CIN excitation

One key observation is that alcohol exposure disrupts CRF-mediated CIN excitation. Both acute alcohol exposure and withdrawal attenuate the CRF-induced increase in CIN activity, indicating that alcohol reduces CIN responsiveness to CRF, a stress-related neuromodulator. Because stress is a well-established trigger for alcohol relapse (*Bertotto et al., 2010*; *Becker et al., 2023*; *Walker et al., 2020*), the observed alcohol-induced disruption of CRF signaling may impair the brain's ability to adaptively respond to stress, thereby increasing susceptibility to relapse. Such dysregulation may compromise stress resilience mechanisms normally mediated by CRF-cholinergic interactions within the striatum and related corticostriatal circuits. Under normal stressful conditions, CRF release onto CINs can modulate cholinergic signaling to adjust behavioral strategies and promote flexible adaptation to environmental demands. However, alcohol-induced impairment of CRF-CIN communication and CIN responsiveness may blunt this adaptive flexibility (*Ma et al., 2022*; *Matamales et al., 2016*). Consequently, stress may instead bias behavior toward habitual or compulsive alcohol seeking, leading to relapse. These findings highlight a potential pathway through which chronic alcohol exposure erodes adaptive stress coping and promotes relapse vulnerability. While these alcohol effects were modest in magnitude, they were consistent across recordings and statistically reliable. It is important to note, however, that ex vivo slice physiology may underestimate alcohol's impact due to factors such as washout, diffusion barriers, and the absence of an intact network state. Thus, future in

vivo studies are needed to evaluate the robustness and behavioral relevance of this modulation. It also remains to be determined whether alcohol pre-exposure alters subsequent CIN responses to ethanol in the absence of CRF, an important question for future studies.

One possibility is that alcohol exposure downregulates CRFR1 expression (*Hansson et al., 2007*; *Zhou et al., 2000*) or alters CIN function (*Ma et al., 2022*; *Huang et al., 2024*), reducing their responsiveness to stress-related input (*Besheer et al., 2014*; *Sayette et al., 2001*). Alcohol may also interfere with intracellular signaling cascades downstream of CRFR1 activation, leading to diminished excitability and ACh release. Another possibility is that alcohol and CRF may converge on CINs through an occlusion mechanism, in which prior activation of CRFR1 reduces the efficacy of subsequent CRF signaling. Although our ex vivo preparation does not readily permit a direct test of this hypothesis, future in vivo approaches with high temporal control will be important to evaluate this possibility.

## Implications for AUD

By delineating a CRF-to-CIN pathway linking the extended amygdala to the dorsal striatum, our findings provide insight into how stress and alcohol may interact at the circuit level to influence striatal processing relevant to behavior. Given the role of CINs in habit learning and behavioral flexibility (*Bradfield et al., 2013*), disruption of this stress-sensitive cholinergic circuit may underlie the decision-making deficits and compulsive alcohol-seeking behavior observed in AUD (*Gangal et al., 2023*; *Ma et al., 2022*; *Huang et al., 2024*; *Zorrilla et al., 2014*; *Moberg et al., 2017*; *Koob and Vendruscolo, 2023*).

In summary, we identify a direct CRF-positive projection from the CeA and BNST to dorsal striatal CINs, revealing a new mechanism by which stress can modulate cholinergic signaling in the striatum. We show that CRF enhances CIN excitability and ACh release, and that alcohol exposure attenuates this modulation. These findings highlight the CRF-CIN circuit as a potential site of vulnerability in AUD and suggest that restoring CRF signaling may help counteract stress and alcohol-induced striatal dysfunction. Future work should assess whether pharmacological or circuit-based interventions targeting CRFR1 or CINs can mitigate behavioral impairments in individuals with AUD.

# Materials and methods
## Animals

Male and female 3- to 4-month-old mice or rats were used in all studies. Rats were mainly used for histological validation of CRFR1 expression with the CRFR1-Cre-tdTomato line, whereas mice were used for histology, electrophysiology, optogenetics, and GRAB-ACh sensor experiments because of the availability of transgenic Cre-driver and reporter lines. ChAT-eGFP (stock 007902), ChAT-Cre (stock 031661), Drd1a-tdTomato (D1-tdT, stock 016204), Ai32 (stock 012569), CRH-ires-CRE (stock 012704), and C57BL/6J (stock 000664) mice were purchased from The Jackson Laboratory (*Tallini et al., 2006*; *Rossi et al., 2011*; *Ade et al., 2011*; *Madisen et al., 2012*; *Taniguchi et al., 2011*). CRFR1-GFP mice were gifted by Dr. Marisa Roberto's lab. All mice were backcrossed onto a C57BL/6 background. ChAT-Cre mice were crossed with D1-tdTomato mice to obtain ChAT-Cre;D1-tdTomato mice. CRF-Cre mice were crossed with ChAT-eGFP mice to generate CRF-Cre;ChAT-eGFP mice. CRF-Cre;Ai32 mice were generated in-house and crossed with ChAT-eGFP to generate triple transgenic CRF-Cre;Ai32;ChAT-eGFP mice. We used CRFR1-Cre-2A-tdTomato rats that have been validated for expression of Cre and tdTomato in CRFR1-expressing neurons (*Weera et al., 2022*). We also used CRF-Cre rats that have been shown to express Cre recombinase in CRF-producing neurons in the CeA and BNST (*Pomrenze et al., 2015*). We crossed CRF-Cre rats with a Cre-dependent tdTomato reporter line to visualize CRF[+] neurons. Genotypes were confirmed through PCR analysis of tail DNA to detect Cre or fluorescent protein genes in mice and rats (Cre for CRF-Cre and ChAT-Cre, tdTomato for D1-tdTomato, and GFP for Ai32) (*Lu et al., 2019*; *Wang et al., 2015a*; *Cheng et al., 2017*; *Cheng et al., 2018*; *Wei et al., 2018*). Animals were randomly assigned to the experimental groups. Animals were housed in a temperature- and humidity-controlled vivarium with a 12 hr light/dark cycle. Food and water were available ad libitum. This study was performed in strict accordance with the recommendations in the Guide for the Care and Use of Laboratory Animals of the National Institutes of Health. All of the animals

were handled according to approved Texas A&M University Institutional Animal Care and Use Committee protocols (approval number: 2022-0198). All surgery was performed under isoflurane anesthesia, and every effort was made to minimize suffering.

## Reagents

AAV8-Ef1a-FLEX-TVA-mCherry (lot # AV5008b), AAV8-FLEX-RG (lot # AV5005f), and AAV-FLEX-Chrimson-tdTomato (lot # AV5844) were purchased from the UNC Vector Core (*Watabe-Uchida et al., 2012*; *Klapoetke et al., 2014*). AAVrg-pCAG-FLEX-eGFP-WPRE (catalog # 51502-AAVrg) was purchased from Addgene (*Oh et al., 2014*), while EnvA-SADΔG-GFP was purchased from the Salk Institute (*Wickersham et al., 2007*). Choline acetyltransferase (AB144P) antibody was purchased from Sigma (*Wang et al., 2015b*). ACh sensor (AAV-GRAB$_{ACh4m}$) was obtained from BrainVTA (PT-7021) (*Purvines et al., 2025*; *Huang et al., 2024*; *Touponse et al., 2025*; *Potjer et al., 2025*; *Gangal et al., 2025*), which is an updated version of ACh3.0 (*Jing et al., 2020*). CRF peptide and NBI 35695 (CRFR1 antagonist) were obtained from Tocris. DNQX, TTX, 4-AP, and bicuculline were also purchased from Tocris.

## Stereotaxic virus infusion

The stereotaxic virus infusion procedure was conducted as described previously (*Lu et al., 2019*; *Ma et al., 2018*; *Roltsch Hellard et al., 2019*). When required for the experimental design, AAV-DIO-TVA-mCherry and AAV-DIO-RG were bilaterally infused into the dorsomedial striatum (AP: 0.38 mm, ML: ±1.55 mm, DV: –2.90 mm from the bregma) (*Gangal et al., 2023*) of ChAT-Cre;D1-tdTomato mice (*Figure 1*). Rabies-GFP was infused at the same injection site 3 weeks later at a 10-degree angle to avoid contamination of the infusion tract. Rabies-GFP virus was allowed to incubate for 1 week. AAVrg-pCAG-FLEX-eGFP-WPRE was infused into the dorsal striatum (AP: 0.00, mm, ML: ±2.80 mm, DV: –4.85 mm from the bregma) (*Ma et al., 2022*; *Huang et al., 2024*) of CRF-Cre;tdTomato rats (*Figure 3A–E*). Additionally, AAV-DIO-Chrimson-tdTomato was infused into the CeA (AP: –1.60 mm; ML: ±4.20 mm; DV: –8.10 mm) and BNST (AP: –0.10 mm; ML: ±1.40 mm; DV: –6.70 mm) of CRF-Cre;ChAT-GFP mice as described in published literature (*de Guglielmo et al., 2019*; *Figure 3F–H*). The animals were placed on a stereotaxic surgical frame after being sedated with 3–4% isoflurane at a rate of 1.0 L/min, as described previously (*Ma et al., 2018*; *Roltsch Hellard et al., 2019*; *Lu et al., 2021*). These coordinates were obtained from previous publications and verified using the rat (*Paxinos et al., 1980*) or mouse (*Franklin and Paxinos, 2007*) brain atlases. A volume of 0.5 µL/site (mice) or 1 µL/site (rats) of virus was infused at a rate of 0.08 µL/min. At the end of the infusion, the injectors remained at the injection site for 10–15 min before removal to allow for virus diffusion. The scalp incision was then sutured, and animals were returned to their home cage for recovery.

## Histology and cell counting

Animals were anesthetized and perfused intracardially with 4% paraformaldehyde (PFA) in phosphate-buffered saline (PBS). The brains were extracted and submerged in 4% PFA/PBS solution for 1 day at 4°C before being transferred to a 30% sucrose solution in PBS. Once the brains completely sank in the sucrose solution, they were cut into 50-µm-thick coronal sections using a cryostat. The slices were stored in a PBS bath at 4°C before mounting on slides for imaging using a confocal laser-scanning microscope (Fluoview, Olympus). All images were processed using Imaris 8.3.1 (Bitplane, Zurich, Switzerland) as previously reported (*Lu et al., 2019*; *Cheng et al., 2017*; *Wei et al., 2018*). Immunostaining for choline acetyltransferase (ChAT) was performed using an anti-ChAT primary antibody (MilliporeSigma; Cat#: AB144P), followed by a fluorophore-conjugated secondary antibody (Thermo Fisher; Cat#: A-21447) emitting at 647 nm to label CINs with far-red fluorescence. Counting was conducted by experimenters blinded to the experiment conditions and group assignments. Cell counting was performed in eight CRF-Cre;tdTomato rats. For each brain region, 5–10 brain sections were imaged from each animal. Imaris was used to quantify green and red neurons, as well as evaluate colocalization. Brain regions were identified using the Mouse Brain Atlas (*Franklin and Paxinos, 2007*). Images were primarily obtained from the posterior dorsomedial striatum, corresponding to coronal slices posterior to the crossing of the anterior commissure and anterior to the tail of the striatum (starting around 0.62 mm and ending at –1.3 mm relative to the bregma).

## Slice electrophysiology

### Slice preparation

Slices were prepared and electrophysiological recordings were conducted as described previously (*Lu et al., 2019*; *Cheng et al., 2021*; *Ma et al., 2018*; *Roltsch Hellard et al., 2019*; *Ma et al., 2017*). Briefly, coronal sections (250 µm) containing the posterior dorsomedial striatum were cut in an ice-cold cutting solution containing (in mM): 40 NaCl, 148.5 sucrose, 4 KCl, 1.25 $NaH_2PO_4$, 25 $NaHCO_3$, 0.5 $CaCl_2$, 7 $MgCl_2$, 10 glucose, 1 sodium ascorbate, 3 sodium pyruvate, and 3 myo-inositol. The solution was saturated with 95% $O_2$ and 5% $CO_2$. Slices were then incubated in a 1:1 mixture of the cutting and external solutions at 32°C for 45 min. The external solution was composed of the following (in mM): 125 NaCl, 4.5 KCl, 2.5 $CaCl_2$, 1.3 $MgCl_2$, 1.25 $NaH_2PO_4$, 25 $NaHCO_3$, 15 glucose, and 15 sucrose. The external solution was saturated with 95% $O_2$ and 5% $CO_2$. Slices were then maintained in the external solution at room temperature until use. Recordings were primarily obtained from the posterior dorsomedial striatum, corresponding to coronal slices posterior to the crossing of the anterior commissure and anterior to the tail of the striatum (starting around 0.62 mm and ending at −1.3 mm relative to the bregma). More anterior slices were occasionally included to increase the sample size.

### Cell-attached recordings

Individual slices were transferred to a recording chamber and continuously perfused with the external solution at 23 mL/min at 32°C. Fluorescent axonal fibers and neurons were visualized using an epifluorescence microscope (Olympus). The presence of tdTomato and GFP allowed cell-type verification, and expression of Ai32 allowed visualization of CRF-producing fibers. CINs in slices were identified by their labeled color, large size, and spontaneous firing. Spontaneous cell-attached CIN firing activity was recorded for 5 min to calculate the average firing frequency. Cells that were not stable during baseline recordings were excluded for further treatments. Light at 470 or 590 nm was delivered from the objective lens for 2 ms at frequencies and durations specified in the figure legends to stimulate channelrhodopsin-expressing axonal fibers in the dorsal striatum. We used a potassium-based intracellular solution containing (in mM): 123 potassium gluconate, 10 HEPES, 0.2 EGTA, 8 NaCl, 2 MgATP, and 0.3 NaGTP, with an osmolarity of ~280 mOsm/L and the pH adjusted to 7.3 using KOH. For protocols involving alcohol application, we pretreated dorsal striatal slices with alcohol (50 mM) for 1 hr outside of the recording chamber. After pretreatment, slices were transferred to the recording chamber, where alcohol was washed out for at least 15 min before recording. The recordings included a 10 min baseline period (without alcohol), followed by 10 min of CRF bath application and 10 min of alcohol exposure. Control slices did not receive alcohol pretreatment but were exposed to acute alcohol following CRF application. In both conditions, the final 10 min alcohol application allowed us to assess whether alcohol pretreatment altered the CIN firing response to subsequent CRF and alcohol exposure. In protocols that required inhibition of glutamatergic and $GABA_A$ receptors, DNQX and bicuculline were used, respectively. Electrophysiology data were acquired using Clampex-10 (Molecular Devices) and analyzed using Clampfit-10 (Molecular Devices) and Mini Analysis (Mini60, Synaptosoft Inc).

## Ex vivo live-tissue confocal imaging of ACh release

Brain slices were kept in a recording chamber perfused with oxygenated external solution (95% $O_2$ and 5% $CO_2$). An Olympus FluoView FV3000 microscope was used with a 10× NA 0.3 and a 40× NA 0.8 water immersion objective, along with a 488 nm and a 561 nm laser. The sample rate of imaging was 2–3 frames per second. Parameters were maintained consistently across all imaging sessions, including laser intensity, HV, gain, offset, and aperture diameter.

## Statistical analysis

Before conducting all experiments shown in this study, we performed a power analysis with SigmaPlot software (12.5, Systat) using the mean and standard deviation from previous studies in our lab (*Wei et al., 2018*; *Gangal et al., 2023*; *Ma et al., 2022*; *Huang et al., 2024*) to determine the required sample sizes to detect a significant difference. Exclusion criteria were pre-established prior to data collection. Animals were excluded from analysis if (1) viral expression was absent or off-target as verified histologically; or (2) electrophysiological recordings did not meet predefined quality criteria (stable baseline and seal). We tested all data for normality before significance testing. If the normality

test failed, we used nonparametric tests, such as the Mann-Whitney U test. All data are expressed as mean ± SEM. Data were analyzed by two-tailed t-test (unpaired or paired), one- or two-way ANOVA with repeated measures, followed by the Tukey or Sidak post hoc test. Significance was determined if $p<0.05$. Statistical analysis was conducted by the SigmaPlot program. Graphs were constructed using the OriginPro (2024b, OriginLab) program.

## Acknowledgements

We thank Yufei Huang for discussions and supplementary data. This research was supported by NIAAA R01AA021505 (JW), R01AA027768 (JW), U01AA025932 (JW), R01MH112768 (NJ), R21AG086907 (NJ) and R01AA026075 (ROM).

## Additional information

### Funding

| Funder | Grant reference number | Author |
| --- | --- | --- |
| National Institute on Alcohol Abuse and Alcoholism | R01AA021505 | Jun Wang |
| National Institute on Alcohol Abuse and Alcoholism | R01AA027768 | Jun Wang |
| National Institute on Alcohol Abuse and Alcoholism | U01AA025932 | Jun Wang |
| National Institute of Mental Health | R01MH112768 | Nicholas J Justice |
| National Institute of Mental Health | R21AG086907 | Nicholas J Justice |
| National Institute on Alcohol Abuse and Alcoholism | R01AA030293 | Jun Wang |
| National Institute on Alcohol Abuse and Alcoholism | R01AA026075 | Robert O Messing |

The funders had no role in study design, data collection and interpretation, or the decision to submit the work for publication.

### Author contributions

Amanda Essoh, Data curation, Formal analysis, Investigation, Visualization, Methodology, Writing – original draft, Writing – review and editing; Xueyi Xie, Data curation, Formal analysis, Investigation, Visualization, Methodology, Writing – review and editing; Himanshu Gangal, Zhenbo Huang, Ruifeng Chen, Ziyi Li, Data curation, Formal analysis, Investigation, Visualization, Methodology; Xuehua Wang, Resources; Valerie Vierkant, Investigation, Writing – original draft; Miguel A Garza, Maria E Secci, Investigation; Lierni Ugartemendia, Resources, Investigation; Nicholas W Gilpin, Robert O Messing, Resources, Supervision, Writing – review and editing; Nicholas J Justice, Resources, Supervision, Funding acquisition, Writing – review and editing; Jun Wang, Conceptualization, Supervision, Funding acquisition, Writing – original draft, Project administration, Writing – review and editing

### Author ORCIDs

Amanda Essoh ⓘ https://orcid.org/0000-0002-7899-5920
Xueyi Xie ⓘ https://orcid.org/0000-0002-2349-1223
Himanshu Gangal ⓘ https://orcid.org/0000-0002-6901-3067
Zhenbo Huang ⓘ https://orcid.org/0000-0003-4857-5500
Ruifeng Chen ⓘ https://orcid.org/0009-0007-4650-959X

Ziyi Li https://orcid.org/0009-0006-6611-669X
Xuehua Wang https://orcid.org/0009-0003-6887-7968
Valerie Vierkant https://orcid.org/0000-0001-7457-497X
Miguel A Garza https://orcid.org/0009-0007-2079-8231
Lierni Ugartemendia https://orcid.org/0000-0001-6423-0188
Maria E Secci https://orcid.org/0000-0001-5769-1250
Nicholas W Gilpin https://orcid.org/0000-0001-8901-8917
Nicholas J Justice https://orcid.org/0000-0002-4673-390X
Robert O Messing https://orcid.org/0000-0002-5345-4431
Jun Wang https://orcid.org/0000-0002-0085-4722

### Ethics

This study was performed in strict accordance with the recommendations in the Guide for the Care and Use of Laboratory Animals of the National Institutes of Health. All of the animals were handled according to approved Texas A&M University Institutional Animal Care and Use Committee protocols (approval number: 2022-0198). All surgery was performed under isoflurane anesthesia, and every effort was made to minimize suffering.

Reviewer #1 (Public review): https://doi.org/10.7554/eLife.107145.3.sa1
Reviewer #2 (Public review): https://doi.org/10.7554/eLife.107145.3.sa2
Reviewer #3 (Public review): https://doi.org/10.7554/eLife.107145.3.sa3
Author response https://doi.org/10.7554/eLife.107145.3.sa4

---

## Additional files

### Supplementary files

MDAR checklist

Source data 1. Source data containing the numerical datasets underlying all figure panels.

### Data availability

Source data are provided with the manuscript in *Source data 1*.

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
