## [Editor Report · eLife Assessment]

This **important** work shows that corticotrophin-releasing factor is delivered monosynaptically to dorsal striatal cholinergic interneurons from the central amygdala and bed nucleus of the stria terminalis. CRF increases cholinergic interneuron firing and release of acetylcholine, and this action is attenuated by pre-exposure to ethanol, suggesting a potential role in stress- and alcohol use disorders. This revision addressed prior concerns, presented **convincing** evidence supporting the conclusions, and set the stage for additional studies.

---

## [Referee Report · Reviewer #1 (Public review)]

Summary:

The authors show that corticotropin-releasing factor (CRF) neurons in the central amygdala (CeA) and bed nucleus of the stria terminalis (BNST) monosynaptically target cholinergic interneurons (CINs) in the dorsal striatum of rodents. Functionally, activation of CRFR1 receptors increases CIN firing rate, and this modulation was reduced by pre-exposure to ethanol. This is an interesting finding, with potential significance for alcohol use disorders.

Strengths:

Well-conceived circuit mapping experiments identify a novel pathway by which the CeA and BNST can modulate dorsal striatal function by controlling cholinergic tone. Important insight into how CRF, a neuropeptide that is important in mediating aspects of stress, affective/motivational processes and drug-seeking, modulates dorsal striatal function.

Weaknesses:

(1) Tracing and expression experiments were performed both in mice and rats (often in non-overlapping ways). While these species are similar in many ways, differences do exist. The authors address this important point in their final text.

(2) As the authors point out, CRF likely modulates CIN activity in both direct and indirect ways. As justified, exploration of the network-level modulation of CINs by CRF (and how these processes may interact with direct modulation via CRFR1 on CINs) is left for future studies.

---

## [Referee Report · Reviewer #2 (Public review)]

Summary:

Essoh and colleagues present a thorough and elegant study identifying the central amygdala and BNST as key sources of CRF input to the dorsal striatum. Using monosynaptic rabies tracing and electrophysiology, they show direct connections to cholinergic interneurons. The study builds on previous findings that CRF increases CIN firing, extending them by measuring acetylcholine levels in slices and applying optogenetic stimulation of CRF+ fibers. It also uncovers a novel interaction between alcohol and CRF signaling in the striatum, likely to spark significant interest and future research.

Strengths:

A key strength is the integration of anatomical and functional approaches to demonstrate these projections and assess their impact on target cells, striatal cholinergic interneurons.

Comments on revisions:

No further concerns or recommendations.

---

## [Referee Report · Reviewer #3 (Public review)]

Summary:

The authors demonstrate that CRF neurons in the extended amygdala form GABAergic synapses on to cholinergic interneurons and that CRF can excite these neurons. The evidence is strong, however the authors lack to make a compelling connection showing CRF released from these extended amygdala neurons is mediating any of these effects. Further, they show that acute alcohol appears to modulate this action, although the effect size is not particularly robust.

Strengths:

This is an exciting connection from the extended amygdala to the striatum that provides a new direction for how these regions can modulate behavior. The work is rigorous and well done.

Weaknesses:

The effects of acute ethanol are modest but consistent, the potential role of this has yet to be determined. Further, the opto stim experiments are conducted in an ai32 mouse, so it is impossible to determine if that is from CEA and BNST, vs. another population of CRF containing neurons. This is an important caveat that was acknowledged.

---

## [Author Response]

The following is the authors’ response to the original reviews

We appreciate the reviewers’ insightful comments. In response, we conducted three new experiments, summarized in Author response table 1. After the table, we provide detailed responses to each comment.

**Author response table 1. sa4table1:** Summary of new experiments and results.

Exp	Aim	Result	Old Fig	New Fig	Concerns Addressed
1	Test whether CRFR1 is expressed in a subset of CINs in mice	~10% CINs express CRFR1 in mice.	4A4C	4D, 4E	Reviewer 1, comment 2
2	Measure the onset latency of opto-evoked CRF+ response on CINs	The average onset latency from the start of stimulation to the start of the response was ~6.5 ms.	3	3J	Reviewer 2, Recommendation 5
3	Test if the CRFR1 antagonist abolishes the optically induced increase in CIN firing	Bath application of CRFR1 antagonist abolished the increase in CIN firing induced by optical stimulation of CRF terminals.	5H	5J, 5K	Reviewer 3, Recommendation 1

**Reviewer #1 (Public review):**
The authors show that corticotropin-releasing factor (CRF) neurons in the central amygdala (CeA) and bed nucleus of the stria terminalis (BNST) monosynaptically target cholinergic interneurons (CINs) in the dorsal striatum of rodents. Functionally, activation of CRFR1 receptors increases CIN firing rate, and this modulation was reduced by pre-exposure to ethanol. This is an interesting finding, with potential significance for alcohol use disorders, but some conclusions could use additional support.Strengths:Well-conceived circuit mapping experiments identify a novel pathway by which the CeA and BNST can modulate dorsal striatal function by controlling cholinergic tone. Important insight into how CRF, a neuropeptide that is important in mediating aspects of stress, affective/motivational processes, and drug-seeking, modulates dorsal striatal function.Weaknesses:(1) Tracing and expression experiments were performed both in mice and rats (in a mostly nonoverlapping way). While these species are similar in many ways, some conclusions are based on assumptions of similarities that the presented data do not directly show. In most cases, this should be addressed in the text (but see point number 2).

In the revised manuscript, we have clarified this limitation in the first paragraph of the Methods and the third paragraph of the Discussion and avoid cross-species claims, limiting our conclusions to the species in which each assay was performed. Specifically, we now state that while mice and rats share many conserved amygdalostriatal components, our tracing and expression studies were performed in a species-specific manner, and direct cross-species comparisons of CRF–CIN connectivity and CRFR1 expression were not assessed. We further note that future studies will be needed to determine the extent to which these observations are conserved across species as more tools become available.

(2) Experiments in rats show that CRFR1 expression is largely confined to a subpopulation of striatal CINs. Is this true in mice, too? Since most electrophysiological experiments are done in various synaptic antagonists and/or TTX, it does not affect the interpretation of those data, but non-CIN expression of CRFR1 could potentially have a large impact on bath CRF-induced acetylcholine release.

To address whether CRFR1 expression in striatal CINs is conserved across species, we performed new histological experiments using CRFR1-GFP mice. Striatal sections were immunostained with anti-ChAT, and we found that approximately 10% of CINs express CRFR1 (new Fig. 4D, 4E). This result indicates that, similar to rats, a subset of CINs in mice express CRFR1. However, the proportion of CRFR1^+^ CINs is lower than the proportion of CRF-responsive CINs observed during electrophysiology experiments, suggesting that CRF may also modulate CIN activity indirectly through network or synaptic mechanisms. We have also noted in the revised Discussion that while CRFR1 expression is confirmed in a subset of CINs, the broader distribution of CRFR1 among other striatal cell types remains to be determined (third paragraph of Discussion).

In our study, bath application of CRF increased striatal ACh release. Because striatal ACh is released primarily from CINs, and CRFR1 is an excitatory receptor, this effect is most likely mediated by CRF activation of CRFR1 on CINs, leading to enhanced CIN activity and ACh release. Although CRFR1 may also be expressed on other striatal neurons, these cell types—medium spiny neurons and GABAergic interneurons—are inhibitory. If CRF were to activate CRFR1 on these GABAergic neurons, the resulting increase in GABA release would suppress CIN activity and consequently reduce, rather than enhance, ACh release. Given that most CINs responded functionally while only a small subset expressed CRFR1, these findings imply that indirect mechanisms, such as CRF modulation of local circuits influencing CIN excitability, may also contribute to the observed increase in ACh release. Together, these data support a model in which CRF primarily enhances ACh release via activation of CRFR1-expressing CINs, while indirect network effects may further amplify this response.

(3) Experiments in rats show that about 30% of CINs express CRFR1 in rats. Did only a similar percentage of CINs in mice respond to bath application of CRF? The effect sizes and error bars in Figure 5 imply that the majority of recorded CINs likely responded. Were exclusion criteria used in these experiments?

We thank the reviewer for this insightful question. In our mouse cell-attached recordings, ~80% of CINs increased firing during CRF bath application, and all recorded cells were included in the analysis (no exclusions based on response direction/magnitude; cells were only required to meet standard recording-quality criteria such as stable baseline firing and seal).

Using a CRFR1-GFP reporter mouse, we found that ~10% of striatal CINs are GFP+, suggesting that the high proportion of CRF-responsive CINs cannot be explained solely by somatic reporter-labeled CRFR1 expression. Importantly, the CRF-induced increase in CIN firing is blocked by the selective CRFR1 antagonist NBI 35695 (Fig. 5B–C), supporting a CRFR1-dependent mechanism at the circuit level. We now discuss several non-mutually exclusive explanations for this apparent discrepancy: (i) reporter lines (e.g., CRFR1-GFP) may underestimate functional CRFR1 expression, particularly for low-level or compartmentalized receptor pools; (ii) bath-applied CRF may act indirectly via CRFR1 on presynaptic afferents, thereby enhancing excitatory drive onto CINs; and (iii) electrical coupling among CINs could allow direct effects in a subset of CINs to propagate through the CIN network (Ren, Liu et al. 2021). We added this discussion to the revised manuscript (fourth paragraph of the Discussion).

(4) The conclusion that prior acute alcohol exposure reduces the ability of subsequent alcohol exposure to suppress CIN activity in the presence of CRF may be a bit overstated. In Figure 6D (no ethanol preexposure), ethanol does not fully suppress CIN firing rate to baseline after CRF exposure. The attenuated effect of CRF on CIN firing rate after ethanol pre-treatment (6E) may just reduce the maximum potential effect that ethanol can have on firing rate after CRF, due to a lowered starting point. It is possible that the lack of significant effect of ethanol after CRF in pre-treated mice is an issue of experimental sensitivity. Related to this point, does pre-treatment with ethanol reduce the later CIN response to acute ethanol application (in the absence of CRF)?

In the revised manuscript, we have tempered our interpretation in the final Results section and throughout the Discussion to emphasize that ethanol pre-exposure attenuates, rather than abolishes, the CRFinduced increase in CIN firing. We also note the reviewer’s important point that in Figure 6D, ethanol does not fully suppress firing to baseline after CRF exposure, consistent with a partial effect. Regarding the reviewer’s question, our experiments were specifically designed to test interactions between CRF and ethanol, so we did not assess whether ethanol pre-treatment alters subsequent responses to ethanol alone. We now explicitly acknowledge CRF-dependent and CRF-independent effects of ethanol on CIN activity as an important point for future studies to disentangle (sixth paragraph of the Discussion). For example, comparing ethanol responses with and without prior ethanol without any treatment with CRF could resolve this question.

(5) More details about the area of the dorsal striatum being examined would be helpful (i.e., a-p axis).

We now provide more detail regarding the anterior–posterior axis of the dorsal striatum examined. Most recordings and imaging were performed in the posterior dorsomedial striatum (pDMS), corresponding to coronal slices posterior to the crossing of the anterior commissure and anterior to the tail of the striatum (starting around 0.62 mm and ending at −1.3 mm relative to the Bregma). While our primary focus was on posterior slices, some anterior slices were included to increase the sample size. These details have been added to the Methods (Last sentence of the ‘Histology and cell counting’ section and of the ‘Slice electrophysiology’ section).

**Reviewer #2 (Public review):**
Essoh and colleagues present a thorough and elegant study identifying the central amygdala and BNST as key sources of CRF input to the dorsal striatum. Using monosynaptic rabies tracing and electrophysiology, they show direct connections to cholinergic interneurons. The study builds on previous findings that CRF increases CIN firing, extending them by measuring acetylcholine levels in slices and applying optogenetic stimulation of CRF+ fibers. It also uncovers a novel interaction between alcohol and CRF signaling in the striatum, likely to spark significant interest and future research.Strengths:A key strength is the integration of anatomical and functional approaches to demonstrate these projections and assess their impact on target cells, striatal cholinergic interneurons.Weaknesses:(1) The nature of the interaction between alcohol and CRF actions on cholinergic neurons remains unclear. Also, further clarification of the ACh sensor used and others is required

We have clarified the nature of the interaction between alcohol and CRF signaling in CINs and have provided additional details regarding the acetylcholine sensor used. These issues are addressed in detail in our responses to the specific comments below.

**Reviewer #2 (Recommendations for the authors):**
(1) The interaction between the effects of alcohol and CRF is a novel and important part of this study. When considering possible mechanisms underlying the findings in the discussion, there is no mention of occlusion. Given that incubation with alcohol produced a similar increase in firing of CINs as CRF, occlusion could be a parsimonious explanation for the observed interaction. Have the author considered blocking the effects of alcohol on CIN with CRF-R1 antagonist? Another experiment that could address the occlusion would be to test if alcohol also increases ACh levels as it did CRF.

We thank the reviewer for proposing occlusion as a potential mechanism underlying the interaction between alcohol and CRF. We agree that, in principle, alcohol-induced endogenous CRF release could occlude subsequent exogenous CRF-mediated potentiation of CIN firing, and we carefully considered this possibility.

However, several observations from our data argue against occlusion driven by acute alcohol exposure or withdrawal in this preparation. First, as shown in Fig. 6A, bath application of alcohol transiently reduced CIN firing, and firing recovered to baseline levels after washout without any rebound increase. Second, in Fig. 6D–E, the baseline firing rates under control conditions and following alcohol pretreatment were comparable, indicating that acute alcohol exposure and short-term withdrawal did not produce a sustained increase in CIN excitability. Together, these results suggest that acute withdrawal in slices is less likely to trigger substantial endogenous CRF release capable of occluding subsequent exogenous CRF effects.

While we and others have previously reported increased spontaneous CIN firing following prolonged in vivo alcohol exposure and extended withdrawal periods (e.g., 21 days), short-term withdrawal (e.g., 1 day) does not robustly alter baseline CIN firing (Ma, Huang et al. 2021, Huang, Chen et al. 2024). Consistent with these prior findings, the absence of a rebound or elevated baseline firing in the present slice experiments discouraged further pursuit of an endogenous CRF occlusion mechanism under acute conditions.

We also considered experimentally testing occlusion by blocking CRFR1 signaling during alcohol pre-treatment. However, this approach is technically challenging in slice recordings, as CRFR1 antagonists require prolonged incubation (~1 hour) during alcohol exposure. Because it is unclear whether endogenous CRF release is triggered by alcohol incubation itself or by withdrawal, the antagonist would need to remain present throughout both the incubation and withdrawal periods. This leaves insufficient time for complete washout of the CRFR1 antagonist prior to subsequent bath application of exogenous CRF to assess its effects on CIN firing. Consequently, residual antagonist presence would confound the interpretation of the exogenous CRF response.

Finally, regarding the possibility that alcohol increases acetylcholine release, we did not observe alcohol-induced increases in CIN firing in slices, arguing against elevated ACh signaling under these conditions. Consistent with prior work (Ma, Huang et al. 2021, Huang, Chen et al. 2024), alcohol-induced increases in CIN excitability and cholinergic signaling appear to depend on prolonged in vivo exposure and extended withdrawal rather than acute slice-level manipulations.

We have now incorporated discussion of occlusion as a potential mechanism (seventh paragraph) and clarified why our data and technical considerations argue against it in the present study. We thank the reviewer for this wonderful suggestion, which we will test in future in vivo studies.

(2) Retrograde monosynaptic tracing of inputs to CIN. Results state the finding of labeling in all previously reported area..." Can the authors report these areas? A list in the text or a bar plot, if there is quantification, will suffice. This formation will serve as important validation and replication of previous findings.

We thank the reviewer for this constructive suggestion. We agree that summarizing the anatomical sources of CIN input provides important validation of our tracing results. In the revised Results, we now list the major input regions observed, including the striatum itself, cortex (e.g., cingulate cortex, motor cortex, somatosensory cortex), thalamus (e.g., parafascicular thalamic nucleus, centrolateral thalamic nucleus), globus pallidus, and midbrain (first paragraph of the Results). Quantitative analysis of relative input strength will be presented in a separate study that expands on these findings. Here, we limit the current manuscript to the functional characterization of CRF and alcohol modulation of CINs.

(3) Given the difference in connectivity among striatal subregions, it would be important to describe in more detail the injection site in the results and figures. In the figure, for example, you might want to include the AP coordinates, given that it is such a zoomed-in image, it is hard to tell how anterior/posterior the site is. I imagine that the picture is a representative image of the injection site, but maybe having a side image with overlay of injection sites in all the animals used, would help.

The anterior–posterior (AP) coordinates for representative images have been included in the panels and reiterated more clearly in the revised Results section and figure legends. In the legend for Figure 3B, a list of AP coordinates for each animal used for Figure 3A-3E has been added.

(4) Figure 1D inset, there seem to be some double-labeled cells in the zoomed in BNST images. The authors might want to comment on this. It seemed far from the injection site. Do D1-MSN so far away show connectivity to CINs?

Upon closer inspection of the BNST images, we noted a small number of double-labeled cells were indeed present, consistent with prior reports that a subset of D1R-expressing neurons (~10%) has been reported previously in our lab in the BNST, with the majority being D2R-expressing neurons (Lu, Cheng et al. 2021). Given the BNST’s anatomical proximity to the dorsal striatum, it is plausible that some D1Rexpressing neurons in this region provide monosynaptic input to CINs, highlighting a potential ventral-to-dorsal connection that merits further study.

(5) Can the author provide quantification of the onset delay of the optogenetic evoked CRF+ axon responses onto CINs? The claim of monosynaptic connectivity is well supported by the TTX/4AP experiment but additional information on the timing will strengthen that conclusion.

We thank the reviewer for this insightful suggestion. Quantifying the onset latency of optogenetically evoked CRFMsup+ axon responses onto CINs provides valuable confirmation of monosynaptic connectivity. To address this, we performed new latency measurements under the same recording conditions as the TTX/4-AP experiments. The average onset latency from the start of the optical stimulation was 5.85 ± 0.37 ms (new Figure 3J), consistent with direct monosynaptic transmission.

As an additional reference, we analyzed latency data from a separate project in which we optogenetically stimulated cholinergic interneurons and recorded synaptic responses in medium spiny neurons. This circuit, known to involve disynaptic transmission from CINs to MSNs via nAChR-expressing interneurons (Autor response image 1) (English, Ibanez-Sandoval et al. 2011), exhibited a significantly longer latency (18.34 ± 0.70 ms; t_(29)_ = 10.3, p < 0.001) compared to CRF⁺ CeA/BNST inputs to CINs (5.85 ± 0.37 ms). Together, these results further support that CRF⁺ axons form direct functional synapses onto CINs.

**Author response image 1. sa4fig1:** Latency of disynaptic transmission from CINs to MSNs via interneurons (A) Schematic illustrating optogenetic stimulation of Chrimson-expressing CINs, leading to excitation of nAChRexpressing interneurons that release GABA onto recorded MSNs. (B) Sample trace of disynaptic transmission (left) and bar graph summarizing onset latency (right) from light stimulation to synaptic response onset (n = 23 neurons from 3 mice).

(6) The ACh sensor reported is "AAV-GRABACh4m" but the reference is for GRAB-ACh3.0. Also, BrainVTA has GRAB-ACh4.3. Is this the vector? Could you please check the name of the construct and report the corresponding reference, as well as clarify the meaning of the additional "m". They have a mutant version of the GRAB-ACH that researchers use for control, and of course, you want to use it as a control, but not for the test experiment.

GRAB-ACh4m is the correct acetylcholine sensor used in this study. The ACh4 series (including ACh4h, ACh4m, and ACh4l; personal communication with Dr. Yulong Li’s lab) represents an updated generation following GRAB-ACh3.0. Although the ACh4 family has not yet been formally published, these constructs are publicly available through BrainVTA (https://www.brainvta.tech/plus/view.php?aid=2680).

The suffix “m” does not indicate a mutant control; rather, it denotes a medium-affinity variant within the ACh4 sensor family. Importantly, the mutant (non-responsive) control sensor is only available for GRAB-ACh3.0 (ACh3.0mut) and does not exist for the ACh4 series.

Our laboratory has previously used GRAB-ACh4m in multiple peer-reviewed publications (Huang, Chen et al. 2024, Gangal, Iannucci et al. 2025, Purvines, Gangal et al. 2025), and its use has also been reported by independent groups in recent preprints (Potjer, Wu et al. 2025, Touponse, Pomrenze et al. 2025). We have now clarified the construct name, its relationship to GRAB-ACh3.0, in the Methods ‘Reagents’ section, and we have corrected the reference accordingly.

(7) Are CRF-R1+ CINs equally abundant in the DMS and DLS? From the image in Figure 4, it seems that a larger percentage of CINs are CRFR1+ in the DLS than in DMS. Is this true? The authors probably already have this data, or it should be easy to get, and it could be additional information that was not studied before.

We did not perform a quantitative comparison of CRFR1+ CIN abundance between the DMS and DLS in the present study. While the representative images in Figure 4 may appear to suggest regional differences, these panels were selected to illustrate labeling quality rather than relative density and should not be interpreted as evidence of unequal distribution. We have clarified this point in the revised Discussion (last sentence of the third paragraph) and note that future studies will be needed to systematically evaluate potential regional differences in CRFR1 expression, which could have important implications for dorsal striatal function.

(8) The manuscript states several times that there are no CRF+ neurons in the dorsal striatum. At the same time, there are reports of the CRF+ neuron in the ventral striatum and its role in learning. Could the authors include mention of the studies by the Lemos group (10.1016/j.biopsych.2024.08.006)

We have revised the Discussion section to clarify that our findings pertain specifically to the dorsal striatum and now acknowledge the presence and functional relevance of CRF+ neurons in the ventral striatum, citing the Lemos group’s study (fifth paragraph of the Discussion).

(9) For the histology analysis, please express cell counts as "density", not just number of cells, by providing an area (e.g., "number of cell/ µm2").

In the revised manuscript, all histological outcomes have been recalculated as cell density (cells/mm^2^) by normalizing raw cell counts to the measured area of each region of interest (ROI). Figures that previously displayed absolute counts now present densities (cells/mm^2^), with corresponding updates made to figure legends and text. We note one exception in Figure 4B, where the comparison between the total number of CINs and CRFR1+ CINs is best represented as cell counts rather than normalized values, as the counting was conducted in the same area (within the same ROI) of the dorsostriatal subregion.

(10) Figure 2C, we can see there are some labeled fibers in the striatum cut. Would it be possible to get a better confocal image?

Figure 2C has been replaced with a higher-quality confocal image captured at the same magnification and scale. The updated image provides improved clarity and resolution, ensuring accurate visualization of labeled CRF+ fibers, but not cell bodies, within the striatum.

(11) The ACh measurements in the slice are very informative and an important addition. I first thought that these experiments with the GRAB-ACh sensor were performed in ChAT-eGFP mice. After reading more carefully, I realized they were done in wild-type mice. Would you include the wildtype label in the figure as well? The ChATeGFP BAC transgenic line was reported to have enhanced ACh packaging and increased ACh release, which could have magnified the signals. So, it is important to highlight the experiments were done in wildtype mice.

We now label with ‘WT mice’ and note in the legend that all GRAB-ACh experiments were performed in wild-type mice, not ChAT-eGFP, to avoid confounds in ACh release. We thank the reviewer for this important suggestion.

**Reviewer #3 (Public review):**
The authors demonstrate that CRF neurons in the extended amygdala form GABAergic synapses onto cholinergic interneurons and that CRF can excite these neurons. The evidence is strong, however, the authors fail to make a compelling connection showing CRF released from these extended amygdala neurons is mediating any of these effects. Further, they show that acute alcohol appears to modulate this action, although the effect size is not particularly robust.Strengths:This is an exciting connection from the extended amygdala to the striatum that provides a new direction for how these regions can modulate behavior. The work is rigorous and well done.Weaknesses:(1) While the authors show that opto stim of these neurons can increase firing, this is not shown to be CRFR1 dependent. In addition, the effects of acute ethanol are not particularly robust or rigorously evaluated. Further, the opto stim experiments are conducted in an Ai32 mouse, so it is impossible to determine if that is from CEA and BNST, vs. another population of CRF-containing neurons. This is an important caveat.

We added recordings with the CRFR1 antagonist antalarmin. Light-evoked increases in CIN firing were abolished under CRFR1 blockade, linking the effect to CRFR1 (Figure 5J, 5K). We also clarify that CRFCre;Ai32 does not isolate CeA versus BNST sources, so we temper regional claims and highlight this as a limitation. The acute ethanol effects are modest but consistent; we expanded the discussion of dose and preparation constraints in acute slice physiology and note that in vivo studies will be needed to define the network-level impact.

**Reviewer #3 (Recommendations for the authors):**
(1) The authors could bring some of this data together by examining CRFR1 dependence of optical stimulationinduced increases in firing. Further, the authors have devoted significant effort to exploring how the BNST and CEA project to the CIN, yet their ephys does not explore site-specific infusion of ChR2 into either region. How are we to be sure it is not some other population of CRF neurons mediating this effect? The alcohol data does not appear particularly robust, but I think if the authors wanted to, they could explore other concentrations. Mostly I think it is important to discuss the limitations of acute alcohol on 5a brain slice.

We thank the reviewer for these thoughtful comments, which helped us strengthen the mechanistic interpretation of the CRF-CIN interaction. In the revised manuscript, we have addressed each point as follows:

- CRFR1 dependence of optogenetically evoked responses: We performed new recordings in which optogenetic stimulation of CRF⁺ terminals in the dorsal striatum was conducted in the presence of the CRFR1 antagonist antalarmin. The increase in CIN firing evoked by light stimulation was abolished under CRFR1 blockade, confirming that this effect is mediated through CRFR1 activation (new Figure 5J, 5K, third paragraph of the corresponding Result section). These results directly link the functional effects of CRF⁺ terminal activation to CRFR1 signaling on CINs.

- CeA vs. BNST projection specificity: The reviewer is correct that CeA and BNST projections were not analyzed separately. As unknown pathways, our experiment was designed to first establish the monosynaptic connections between CeA/BNST CRF neurons to striatal CINs. Future studies would further explore the specific contribution of each site. However, our data exclude the possibility of other CRF neurons as we selectively infused Cre-dependent opsins into both CeA and BNST of CRF-Cre mice (Figure 3G-3J).

- Limitations of acute slice experiments: We have expanded the Discussion (sixth paragraph) to acknowledge that acute slice physiology cannot fully capture the dynamic and network-level effects of ethanol observed in vivo. While this preparation enables mechanistic precision, factors such as washout, diffusion constraints, and the absence of systemic feedback may underestimate ethanol’s impact on CINs. We now explicitly note this limitation and highlight the need for in vivo studies to examine behavioral and circuit-level implications of CRF–alcohol interactions.

Collectively, these revisions clarify the CRFR1 dependence of CRF^+^ terminal effects and reaffirm that both CeA and BNST projections contribute to CIN modulation while addressing the methodological limitations of the slice preparation.

**Reviewer #4 Public Review:**
This manuscript presents a compelling and methodologically rigorous investigation into how corticotropin-releasing factor (CRF) modulates cholinergic interneurons (CINs) in the dorsal striatum - a brain region central to cognitive flexibility and action selection-and how this circuit is disrupted by alcohol exposure. Through an integrated series of anatomical, optogenetic, electrophysiological, and imaging experiments, the authors uncover a previously uncharacterized CRF⁺ projection from the central amygdala (CeA) and bed nucleus of the stria terminalis (BNST) to dorsal striatal CINs.Strengths:Key strengths of the study include the use of state-of-the-art monosynaptic rabies tracing, CRF-Cre transgenic models, CRFR1 reporter lines, and functional validation of synaptic connectivity and neurotransmitter release. The finding that CRF enhances CIN excitability and acetylcholine (ACh) release via CRFR1, and that this effect is attenuated by acute alcohol exposure and withdrawal, provides important mechanistic insight into how stress and alcohol interact to impair striatal function. These results position CRF signaling in CINs as a novel contributor to alcohol use disorder (AUD) pathophysiology, with implications for relapse vulnerability and cognitive inflexibility associated with chronic alcohol intake. The study is well-structured, with a clear rationale, thorough methodology, and logical progression of results. The discussion effectively contextualizes the findings within broader addiction neuroscience literature and suggests meaningful future directions, including therapeutic targeting of CRFR1 signaling in the dorsal striatum.Weaknesses:(1) Minor areas for improvement include occasional redundancy in phrasing, slightly overlong descriptions in the abstract and significance sections, and a need for more concise language in some places. Nevertheless, these do not detract from the manuscript's overall quality or impact. Overall, this is a highly valuable contribution to the fields of addiction neuroscience and striatal circuit function, offering novel insights into stress-alcohol interactions at the cellular and circuit level, which requires minor editorial revisions.

We have streamlined the abstract and significance statement, reduced redundancy, and improved conciseness throughout the text. We appreciate the reviewer’s feedback, which has helped us further strengthen the clarity and readability of the manuscript.

**Reviewer #4 (Recommendations for the authors):**
(1) Line 29-30: Slightly verbose. Consider: "Alcohol relapse is associated with corticotropin-releasing factor (CRF) signaling and altered reward pathway function, though the precise mechanisms are unclear."

The sentence has been revised as recommended to improve clarity and conciseness in the introductory section (Lines 31-32).

(2) Lines 39-43: Good synthesis, but could better emphasize the novelty of identifying a CRF-CIN pathway.

The abstract has been revised to more clearly emphasize the novelty of identifying a CRF-CIN pathway and its functional significance (Line 42-43).

(3) Lines 66-68: Consider integrating clinical relevance more directly, e.g., "AUD affects over 14 million adults in the U.S., with relapse often triggered by stress...".

The introduction has been revised to more directly emphasize the clinical relevance of alcohol use disorder, including its high prevalence and the role of stress in relapse, thereby underscoring the translational significance of our findings (Lines 68-69).

(4) Line 83: Repetition of "goal-directed learning, habit formation, and behavioral flexibility" appears multiple times; consider variety.

We have varied the phrasing in the Introduction to avoid redundancy. Specifically, in place of repeating “goal-directed learning, habit formation, and behavioral flexibility,” we now use alternative terms such as “action selection,” “habitual responding,” and “cognitive flexibility,” depending on the context.

(5) Lines 107-116: Clarify why both rats and mice were used-do they serve different experimental purposes?

We now explain that each species was used for complementary experimental purposes. Rats were used for histological validation of CRFR1 expression using the CRFR1-Cre-tdTomato line, which has been extensively characterized in this species. Mice were used for the majority of electrophysiological, optogenetic, and GRAB-ACh sensor experiments due to the availability of well-established transgenic CRF-Cre-driver lines. This division allowed us to leverage the most appropriate tools in each species to address different aspects of the study. We have clarified this rationale in the Methods (first paragraph of the “Animals” section) and Discussion (third paragraph).

(6) Electrophysiology section: The distinction between acute exposure vs. withdrawal could be further emphasized.

To better highlight the distinction between acute alcohol exposure and withdrawal, we have clarified the timing and context of each condition within the Results section for Figure 6. Specifically, we now distinguish the immediate suppressive effects of alcohol observed during bath application (acute exposure) from the subsequent changes in CIN firing measured after washout (withdrawal). These revisions clarify the temporal dynamics and functional implications of CRF–alcohol interactions in our experimental design.

(7) Lines 227-229: Reword for clarity: "Significantly more BNST neurons projected to CINs compared to the CeA...".

The sentence has been reworded to clarify as recommended (Lines 247-248).

(8) Lines 373-374: Consider connecting the CRF-CIN circuit to behavioral inflexibility in AUD more directly.

We have modified the sentence (Lines 390-395) to more explicitly link alcohol-induced dysregulation of the CRF–CIN circuit to behavioral inflexibility in AUD, consistent with the established role of CINs in action selection and cognitive flexibility.

(9) Lines 387-389: This is an excellent point about stress resilience; consider expanding with examples or potential implications.

We thank the reviewer for this insightful suggestion. In the revised Discussion (sixth paragraph), we expanded this section to more directly connect alcohol-induced disruption of CRF–CIN signaling with impaired stress resilience and behavioral inflexibility. Specifically, we now note that such dysregulation may compromise stress resilience mechanisms mediated by CRF–cholinergic interactions in the striatum and related corticostriatal circuits. We further discuss how impaired CIN responsiveness could blunt adaptive behavioral adjustments under stress, biasing animals toward habitual or compulsive alcohol seeking. This addition highlights the broader implication that alcohol-induced alterations in CRF–CIN signaling may contribute to relapse vulnerability by undermining adaptive stress coping.

References

English, D. F., O. Ibanez-Sandoval, E. Stark, F. Tecuapetla, G. Buzsaki, K. Deisseroth, J. M. Tepper and T. Koos (2011). "GABAergic circuits mediate the reinforcement-related signals of striatal cholinergic interneurons." Nat Neurosci 15(1): 123–130.

Gangal, H., J. Iannucci, Y. Huang, R. Chen, W. Purvines, W. T. Davis, A. Rivera, G. Johnson, X. Xie, S. Mukherjee, V. Vierkant, K. Mims, K. O'Neill, X. Wang, L. A. Shapiro and J. Wang (2025). "Traumatic brain injury exacerbates alcohol consumption and neuroinflammation with decline in cognition and cholinergic activity." Transl Psychiatry 15(1): 403.

Huang, Z., R. Chen, M. Ho, X. Xie, H. Gangal, X. Wang and J. Wang (2024). "Dynamic responses of striatal cholinergic interneurons control behavioral flexibility." Sci Adv 10(51): eadn2446.

Lu, J. Y., Y. F. Cheng, X. Y. Xie, K. Woodson, J. Bonifacio, E. Disney, B. Barbee, X. H. Wang, M. Zaidi and J. Wang (2021). "Whole-Brain Mapping of Direct Inputs to Dopamine D1 and D2 Receptor-Expressing Medium Spiny Neurons in the Posterior Dorsomedial Striatum." Eneuro 8(1).

Ma, T., Z. Huang, X. Xie, Y. Cheng, X. Zhuang, M. J. Childs, H. Gangal, X. Wang, L. N. Smith, R. J. Smith, Y. Zhou and J. Wang (2021). "Chronic alcohol drinking persistently suppresses thalamostriatal excitation of cholinergic neurons to impair cognitive flexibility." J Clin Invest 132(4): e154969.

Potjer, E. V., X. Wu, A. N. Kane and J. G. Parker (2025). "Parkinsonian striatal acetylcholine dynamics are refractory to L-DOPA treatment." bioRxiv.

Purvines, W., H. Gangal, X. Xie, J. Ramos, X. Wang, R. Miranda and J. Wang (2025). "Perinatal and prenatal alcohol exposure impairs striatal cholinergic function and cognitive flexibility in adult offspring." Neuropharmacology 279: 110627.

Ren, Y., Y. Liu and M. Luo (2021). "Gap Junctions Between Striatal D1 Neurons and Cholinergic Interneurons." Front Cell Neurosci 15: 674399.

Touponse, G. C., M. B. Pomrenze, T. Yassine, V. Mehta, N. Denomme, Z. Zhang, R. C. Malenka and N. Eshel (2025). "Cholinergic modulation of dopamine release drives effortful behavior." bioRxiv.